# SO₂ emissions derived from TROPOMI observations over India using a flux-divergence method with variable lifetimes

Yutao Chen[1,2], Ronald J. van der A[1], Jieying Ding[1], Henk Eskes[1], Jason E. Williams[1], Nicolas Theys[3], Athanasios Tsikerdekis[1], Pieternel F. Levelt[1,2,4]

[1]Royal Netherlands Meteorological Institute (KNMI), De Bilt, the Netherlands
[2]Department of Geoscience & Remote Sensing, Delft University of Technology (TUD), Delft, the Netherlands
[3]Royal Belgian Institute for Space Aeronomy (BIRA-IASB), Brussels, Belgium
[4]National Center for Atmospheric Research (NCAR), Boulder, Colorado, the United States

*Correspondence to*: Yutao Chen (yutao.chen@knmi.nl), Jieying Ding (jieying.ding@knmi.nl)

**Abstract.** The rapid development of the economy and the implementation of environmental policies adapted in India has led to fast changes of regional SO₂ emissions. We present a monthly SO₂ emission inventory for India covering December 2018 to November 2023 based on the TROPOMI Level-2 COBRA SO₂ dataset, by using an improved flux-divergence method and estimated local SO₂ lifetime which includes both its chemical loss and dry deposition. We update the methodology to use the daily CAMS model output estimates of the hydroxyl-radical distribution as well as the measured dry deposition velocity to account for the variability in the tropospheric SO₂ lifetime. It is the first effort to derive the local SO₂ lifetime for application in the divergence method. The results show the application of the local SO₂ lifetime improves the accuracy of SO₂ emissions estimation when compared to calculations using a constant lifetime. Our improved flux-divergence method reduced the spreading of the point source emissions compared to the standard flux-divergence method. Our derived averaged SO₂ emissions covering the recent 5 years are about 5.2 Tg year⁻¹ with a monthly mean uncertainty of 40%, which is lower than the bottom-up emissions of 11.0 Tg year⁻¹ from CAMS-GLOB-ANT v5.3. The total emissions from the 92 largest point source emissions are estimated to be 2.9 Tg year⁻¹, lower than the estimation of 5.2 Tg year⁻¹ from the global SO₂ catalog MSAQSO2LV4. We claim that the variability in the SO₂ lifetime is important to account for in estimating top-down SO₂ emissions.

**1 Introduction**

Sulfur dioxide ($SO_2$) is a reactive gas-phase air pollutant released through natural processes, such as volcanic eruptions and passive degassing (Oppenheimer et al., 2011; Carn et al., 2017), as well as anthropogenic activities, primarily from thermal power plants, fossil fuel combustion, and metal smelting and refining (Smith et al., 2011; Klimont et al., 2013; Serbula et al., 2014). After being released into the atmosphere, $SO_2$ is primarily oxidized in the gas-phase by the hydroxyl radical (OH) to form sulfuric acid ($H_2SO_4(g)$) or scavenged into cloud droplets and subsequential oxidized to form sulphate ($SO_4^{2-}$) via the reaction of ozone and hydrogen peroxide (Steinfeld, 1998). Gaseous $SO_2$ and particulate $SO_4^{2-}$ have detrimental effects on human health via increasing the Particulate Matter concentrations (PM1.0, PM2.5). Exposure to $SO_2$ pollution, whether long or short term, is associated with increased respiratory morbidity (Chen et al., 2007; Clark Nina et al., 2010; Chen et al., 2012; Rodriguez-Villamizar et al., 2015). Sulfuric acid rain induces acidification in both aquatic and terrestrial ecosystems, causing harm to animals and plants (Larssen et al., 2006; Shukla et al., 2013). Additionally, $SO_4^{2-}$ contributes to reduced visibility (Leaderer et al., 1979) and acts as a precursor of cloud formation via increasing the Cloud Condensation Nuclei (CCN), subsequently impacting regional and global climate (Lelieveld and Heintzenberg, 1992; Arnold, 2006).

There have been profound changes regarding global anthropogenic $SO_2$ emissions in the past decades. Specifically, global $SO_2$ emissions have decreased by 31% between 1990-2015 due to the mitigation efforts in Europe and the USA, which have reduced regional $SO_2$ emissions, while East Asia witnessed a 70% increase in 1990-2005, followed by a decreasing trend thereafter (Kuttippurath et al., 2022). Contrary to the declining trend in China (Klimont et al., 2013; Li et al., 2017b; Zheng et al., 2018; van der A et al., 2017; Qu et al., 2019), Indian emissions have surged from 4.5 to 15.0 TgS per year between 1990 and 2015 (Crippa et al., 2018; Aas et al., 2019), after which India became the world's largest emitter of anthropogenic $SO_2$. (Li et al., 2017b; Li et al., 2017a). Given India's substantial dependence on coal-based thermal power plants to fulfill its growing energy demand, it is anticipated that the emissions will continue to rise driven by the population growth and economic development (Venkataraman et al., 2018).

With the development of satellite-based measuring instruments, not only the large $SO_2$ sources, but also the weaker ones, can be monitored from space. These satellite measurements provide effective near real-time information, including $SO_2$ Vertical Column Densities (VCDs), data quality (QA value), to locate the potential $SO_2$ hot spots and estimate point-source emission terms. During the 1980s, only $SO_2$ emitted from large volcano eruptions could be monitored from space by the Total Ozone Mapping Spectrometer (TOMS) and the Solar Backscattered Ultraviolet (SBUV) instruments (Krueger, 1983; McPeters et al., 1984; Krueger et al., 2000). After that, the Global Ozone Monitoring Experiment (GOME), launched in 1995, enabled the detection of large industrial $SO_2$ sources for the first time (Eisinger and Burrows, 1998; Khokhar et al., 2008). Subsequently, the SCanning Imaging Absorption spectroMeter for Atmospheric CHartographY (SCIAMACHY) instrument launched in 2002 (Bovensmann et al., 1999), the Global Ozone Monitoring Experiment-2 (GOME 2) instrument launched in 2006 (Callies et al., 2000), and the Dutch-Finnish Ozone Monitoring Instrument (OMI) instrument launched in 2004 (Levelt et al., 2006) were used to detect sources and monitor emissions from human activities with greater details (Carn et al., 2007; Lee et al., 2011; McLinden et al., 2016). Half of the reported anthropogenic sources can be detected and quantified with OMI $SO_2$ measurements (Fioletov et al., 2015; 2016). Nowadays, the

Tropospheric Monitoring Instrument (TROPOMI) on the ESA Copernicus Sentinel-5P satellite has become one of the most widely used satellite-based monitoring instruments (Veefkind et al., 2012; Theys et al., 2017). TROPOMI supplies daily global coverage for $SO_2$ Tropospheric Vertical Column Densities (TVCDs) from 2018 to the present. The measurements have a horizontal resolution of approximately 5.5 km × 3.5 km (7 km × 3.5 km before August 6, 2019) at nadir viewing geometry. The TROPOMI $SO_2$ product reprocessed by the Covariance-Based Retrieval Algorithm (COBRA) has largely reduced the $SO_2$ noise level and uncertainties as compared to earlier $SO_2$ datasets derived from TROPOMI or other satellite instruments (Theys et al., 2021). It makes the $SO_2$ measurements more sensitive to minor $SO_2$ sources down to 8.0 Gg year$^{-1}$(Theys et al., 2021), which indicates that more $SO_2$ sources can be detected and quantified with the COBRA datasets (Fioletov et al., 2023).

With the significant advancement of satellite-based monitoring instruments over the past decades, a variety of inversion methods have been developed to constrain emissions more efficiently. Data assimilation has been used by combining satellite observations and a chemical transport model (CTM) to derive emissions of trace gases, such as $NO_x$ (Miyazaki et al., 2017; Mijling and van der A, 2012), VOCs (Koohkan et al., 2013), $CH_4$ (Meirink et al., 2008) and $SO_2$ (Tsikerdekis et al., 2023). The mass balance method is a less expensive approach for deriving emissions directly from satellite observations without involving a CTM. For example, Leue et al. (2001) and Martin et al. (2003) started to calculate the $NO_x$ emissions based solely on sink terms, ignoring the effect of atmospheric transport. Recently, Fioletov et al. (2011; 2015; 2016; 2023) identified $SO_2$ point sources using a plume fitting method and quantified emissions based on the mass balance principle with a fixed 6-hour effective time. Beirle et al. (2011) used the plume fitting method to derive the $NO_x$ emissions from the large megacity sources and a mean lifetime of $NO_2$ of 4 hours. Later, Beirle et al. (2019) determined the total $NO_x$ emissions using the divergence method, while also calculating point-source emissions using a 2D-Gaussian peak fitting method with a fixed 4-hour lifetime. It is noteworthy that the sink term, controlled by the tropospheric lifetime, plays a crucial role in determining the final emission terms according to the mass balance principle. However, previous studies have assumed a constant lifetime for the sink term estimation, which can lead to the spreading of emissions (Beirle et al., 2019). Consequently, deriving realistic local $SO_2$ lifetimes, which varies from several hours to several days (Chin et al., 2000; Hains et al., 2008; Lee et al., 2011; Green et al., 2019), is crucial to calculate quantitatively accurate $SO_2$ emissions.

In this study, we will constrain Indian $SO_2$ emissions for the period December 2018 to November 2023 based on daily TROPOMI $SO_2$ observations. The flux-divergence method, i.e. combining the independently derived $SO_2$ sink and divergence, is used to obtain local emissions. Since the sink term is determined by the lifetime, we will initially derive the $SO_2$ local effective lifetime by incorporating the $SO_2$ chemical loss and dry deposition. Subsequently, we will improve the divergence method to generate a high resolution of 0.1° × 0.1° emission map, mitigating the smoothing of the emission map. We will estimate the $SO_2$ emissions using the derived $SO_2$ local lifetimes and the enhanced divergence method. We then will conduct a comparative analysis with existing bottom-up and top-down emission data. The article is organized as follows: the datasets for the divergence and sink terms calculation, and $SO_2$ emissions datasets against which our results are compared are introduced in Sect. 2. Section 3 discusses the basic principles of emission calculation and the method to derive the $SO_2$ lifetimes. Section 4 illustrates the magnitude of the spreading in the original divergence method and how we reduce this smoothing of the emission map on various spatial resolutions. The uncertainties associated with the resulting $SO_2$ emission estimates are discussed in Sect. 5. The regional Indian emission estimations, comparisons with respect to existing

estimates, and emission changes during the study period are given in Sect. 6. Section 7 discusses the uncertainties which are difficult to quantify. Finally, in Section 8 we present our conclusions.

## 2 Data

### 2.1 Satellite observations and wind field datasets

TROPOMI on the ESA Copernicus Sentinel-5P satellite was launched on 13 October 2017 (Veefkind et al., 2012). TROPOMI is a hyperspectral nadir sensor consisting of UV–Vis–NIR spectrometers, monitoring key atmospheric species with high accuracy, including $NO_2$ $O_3$, $SO_2$, $CH_4$, CO, and HCHO as well as aerosol and cloud information. The Sentinel-5P satellite overpass time is about 13:30 local time. The spatial resolution for the center of the swath is approximately 5.5 km × 3.5 km (7 km × 3.5 km before August 6, 2019) in nadir, and 5.5 km × 6 km on average over the swath. In this study, the $SO_2$ emissions are based on the TROPOMI $SO_2$ product reprocessed by the Covariance-Based Retrieval Algorithm (COBRA) (Theys et al., 2021). The TROPOMI Level-2 COBRA $SO_2$ data v01.00.01 is extracted from December 1, 2018 to November 30, 2023 for the $SO_2$ divergence calculation. To ensure the high quality of the measurements, only data with a "QA value" larger than 0.5 and "surface height" lower than 3 km are used. (https://data-portal.s5p-pal.com/product-docs/so2cbr/S5P-BIRA-PRF-SO2CBR_1.0.pdf, last access: 29 July, 2024). Wind field information is needed for the divergence calculation. We used the wind field from the daily operational 12h forecasts of European Centre for Medium-range Weather Forecasts (ECMWF) with a resolution of 0.25° × 0.25° (https://www.ecmwf.int/en/forecasts, last access: 29 July, 2024). The wind fields are interpolated at the mid-point of the Planetary Boundary Layer (PBL).

### 2.2 Copernicus Atmospheric Monitoring Service (CAMS) datasets

CAMS have been regularly publishing global forecasts for atmospheric composition from 2015 to present on the ECMWF website (https://ads.atmosphere.copernicus.eu, last access: 29 July, 2024) (referred to as the CAMS forecast datasets hereafter). The forecast itself uses ECMWF's Integrated Forecast System (IFS) for the data assimilation and modeling of the concentration of over 50 chemical species (including $SO_2$ and OH), 7 different types of aerosols, and several meteorological factors provided with a resolution of 0.4° × 0.4°. The CAMS forecast datasets are available for 137 vertical layers with a temporal resolution of 3 hours.

Calculating the chemical lifetimes of $SO_2$ involves deriving a monthly mean OH climatology (derived from 5-year OH concentration as detailed in Section 3.2). This climatology is based on the monthly mean OH concentrations accessible within the CAMS forecast datasets. Specifically, the OH concentration averaged within the PBL at 6 UTC (11:30AM local time) are used. To ensure a stable OH climatology less influenced by extreme weather events, such as large-scale precipitation occurring on individual days, the monthly mean OH concentrations are averaged over the years from 2018 to 2023. Since cloud cover has minimal influence on the OH concentration over India (Duncan et al., 2024), we did not apply any filtering on OH data based on cloud fraction when computing the OH climatology.

### 2.3 $SO_2$ emission and source datasets

Indian $SO_2$ emissions taken from the bottom-up inventories, i.e. the Emissions Database for Global Atmospheric Research version 8 (EDGARv8) between 2018 to 2022 (Crippa et al., 2024), CAMS global anthropogenic monthly

emissions version 5.3 (CAMS-GLOB-ANTv5.3) (Soulie et al., 2023) from 2018 to 2023, and the top-down $SO_2$ global catalog, the Multi-Satellite Air Quality Sulfur Dioxide ($SO_2$) database Long-Term L4 Global V2 (refers to MSAQSO2L4 hereafter) (Fioletov et al., 2023) from 2019 to 2022, are used for the comparison of the final emission fluxes.. The total Indian emissions from CAMS-GLOB-ANT v5.3 and EDGAR v8 show little variation in recent years, about 11.0 Tg year$^{-1}$ in each year. The total emissions of India's 92 large point sources from MSAQSO2LV4 are 5.3, 4.9, 5.2, 5.6 Tg year$^{-1}$ during 2019 to 2022, respectively. The locations of Indian thermal power plants we use in this study originates from the Open Infrastructure Map (https://openinframap.org/stats/area/India, last access: 29 July, 2024).

## 3 Method description

This flux-divergence method is initially proposed by Beirle et al. (2019) and has been refined and applied in estimating emissions of trace-gases like $NO_x$ (Beirle et al., 2021) and $CH_4$ (Liu et al., 2021). Here we apply it for the derivation of $SO_2$ emissions. The steady-state equation governing the flux-divergence method is described as follows:

$$E = D + S, \tag{1}$$

with $D$, $E$ and $S$ being the terms of divergence, emission and sink of $SO_2$, respectively. This equation shows that the $SO_2$ emissions are obtained by adding estimates of $SO_2$ divergence and sink terms. The two main steps, the divergence calculation, and the sink calculation, are discussed below.

### 3.1 Calculation of the divergence

Eq. (2) defines divergence ($D$) as the continuity equation of the flux ($\vec{F}$), incorporating $SO_2$ VCDs ($V$) and wind field data ($\vec{w}$):

$$D = \nabla \cdot \vec{F} = \nabla(\vec{w} \cdot V). \tag{2}$$

Note that because both VCDs and wind information are available on a grid-scale rather than a continuous state, the Second Order Central Finite Difference Method (SOCFDM) is used to approximate the divergence. The daily divergence of a grid cell needs to be derived for both $x$ and $y$ directions (see the one-dimensional example in supplementary information).

### 3.2 Calculation of the sink term

The relation between sink term, atmospheric density, and lifetime can be expressed as:

$$S = \frac{V_{so_2}}{\tau}, \tag{3}$$

with $S$ the $SO_2$ sink term, $V_{so_2}$ the $SO_2$ VCD, and $\tau$ the $SO_2$ effective lifetime. The $SO_2$ VCDs are taken from the satellite measurements. The $SO_2$ lifetime is determined by various processes in the atmosphere responsible for removing $SO_2$, including deposition and chemical loss. As the deposition and chemical loss occurs simultaneously, the $SO_2$ effective lifetime $\tau$ is defined as follows:

$$\frac{1}{\tau} = \frac{1}{\tau_c} + \frac{1}{\tau_d}, \tag{4}$$

where $\tau_c$ is the chemical lifetime and $\tau_d$ is the lifetime related to $SO_2$ dry deposition.

### 3.2.1 Calculation of the chemical lifetime

Notably, TROPOMI can "see" $SO_2$ only in the cloud-free part of the pixel, leaving $SO_2$ concentrations within or beneath clouds being unmeasurable. We assume that the resulting $SO_2$ has had no interaction with clouds, thus the resulting lifetime derived for $SO_2$ pertains to cloud-free conditions in a constrained region. Oxidization by OH(g) determines the $SO_2$ chemical lifetime in the atmosphere under cloud-free conditions (Blitz et al., 2003; Long et al., 2017; Green et al., 2019). This reaction occurs primarily during daytime hours with maximum sunlight

under humid conditions. Considering the TROPOMI overpass time is 1:00PM local time, coinciding with peak OH concentrations and favorable conditions for $SO_2$+OH reaction, we assume the $SO_2$ lifetime dominance via OH oxidation. Therefore, we use the model simulated OH concentration at 11:30AM local time, which is closest to the TROPOMI overpass time from CAMS forecast datasets, to calculate the chemical lifetime $\tau_c$ (s$^{-1}$) as follows:

$$\tau_c = \frac{1}{k[OH]},\qquad\qquad\qquad\qquad\qquad\qquad\qquad\qquad\qquad (5)$$

with $k$ being the chemical rate coefficient (molecules$^{-1}$ cm$^3$ s$^{-1}$) and [$OH$] denoting the OH concentration (molecules cm$^{-3}$), i.e., OH column density within PBL divided by the PBL height. The rate coefficient $k$ depends on the atmospheric temperature, and is calculated following Table2-1 in Vladimir et al. (2015). Due to the OH concentrations exhibiting a clear seasonal cycle (Lelieveld et al., 2016), we derive a monthly OH climatology

(December 2018 to November 2023) and calculate $k$ to estimate the $SO_2$ chemical lifetime per month per grid cell as shown in Fig. S1. The estimated $SO_2$ monthly mean chemical lifetime varies from 16 to 34 hours. While the distribution of the $SO_2$ chemical lifetime does not show big differences within the same season, it has a clear seasonality, with the lowest chemical lifetime occurring in summer and the highest in winter. The chemical lifetimes averaged for the whole of India in winter, spring, summer, and autumn are 31, 18, 16 and 22 hours,

respectively. The variation in $SO_2$ chemical lifetime is also notable across various regions. The $SO_2$ chemical lifetime in northern regions is larger than that in the south, with an exception occurring in summer when there is less spatial variation in lifetime. This is because more OH can be generated at low latitudes in the lower to middle troposphere due to the small solar zenith angle and high concentration of water vapor (Crutzen and Zimmermann, 1991; Spivakovsky et al., 2000). As these papers show the OH concentration near the Equator remains consistently

high throughout all seasons, leading to less variable chemical lifetimes in southern India compared to the north (Fig. S1).

### 3.2.2 Deposition lifetime

Wet and dry deposition influences the $SO_2$ lifetime in the atmosphere. However, given that all $SO_2$ are measured in cloud-free areas, our analysis only considers the impact of dry deposition i.e. direct loss to the surface. Previous

studies have indicated that $SO_2$ dry deposition lifetimes spanning several days (Matsuda et al., 2006; Faloona et al., 2009; Hayden et al., 2021). Here we use an 0.4 cm s$^{-1}$ as a general dry deposition velocity, which is based on measurements from Hicks (2006), Myles et al. (2007) and Faloona et al. (2009). The $SO_2$ monthly dry deposition lifetime within the PBL height is calculated by dividing the PBL height (from ECMWF data) by 0.4 cm s$^{-1}$ (Slinn et al., 1978). As shown in Fig. S2, the Indian $SO_2$ monthly mean dry deposition lifetime varies from 55 to 135

hours, with the longest lifetime occurring in spring. The dry deposition lifetimes averaged over the whole of India

in winter, spring, summer, and autumn are 62, 120, 75, and 70 hours, respectively. The lifetime is longer in spring due to the higher PBL in this season.

### 3.2.3 The SO$_2$ effective lifetime

Following Eq. (4) we combine the SO$_2$ chemical lifetime and dry deposition terms to calculate the SO$_2$ monthly effective lifetime for each grid-cell to derive the local sink term. The SO$_2$ monthly mean effective lifetime in India varies from 12 to 19 hours (Fig. S3). Figure 1 displays SO$_2$ effective lifetimes averaged for each season. The SO$_2$ seasonal mean lifetimes averaged for India in winter, spring, summer, and autumn are 19, 15, 12, and 16 hours, respectively. After considering the SO$_2$ dry deposition, the annual mean SO$_2$ effective lifetime decreases by 27% compared to only considering the chemical loss, reducing the fraction transported away from strong point sources. Comparing the SO$_2$ lifetime derived here with those proposed in the literature shows that our estimates are similar to other independent model-based lifetime estimates (Lee et al., 2011) and ground-measurement based lifetime estimates (Hains et al., 2008). We therefore argue that on average our calculated SO$_2$ lifetime is reasonable. Furthermore, it has a latitude and seasonal dependency that is often lacking in other inversion methods.

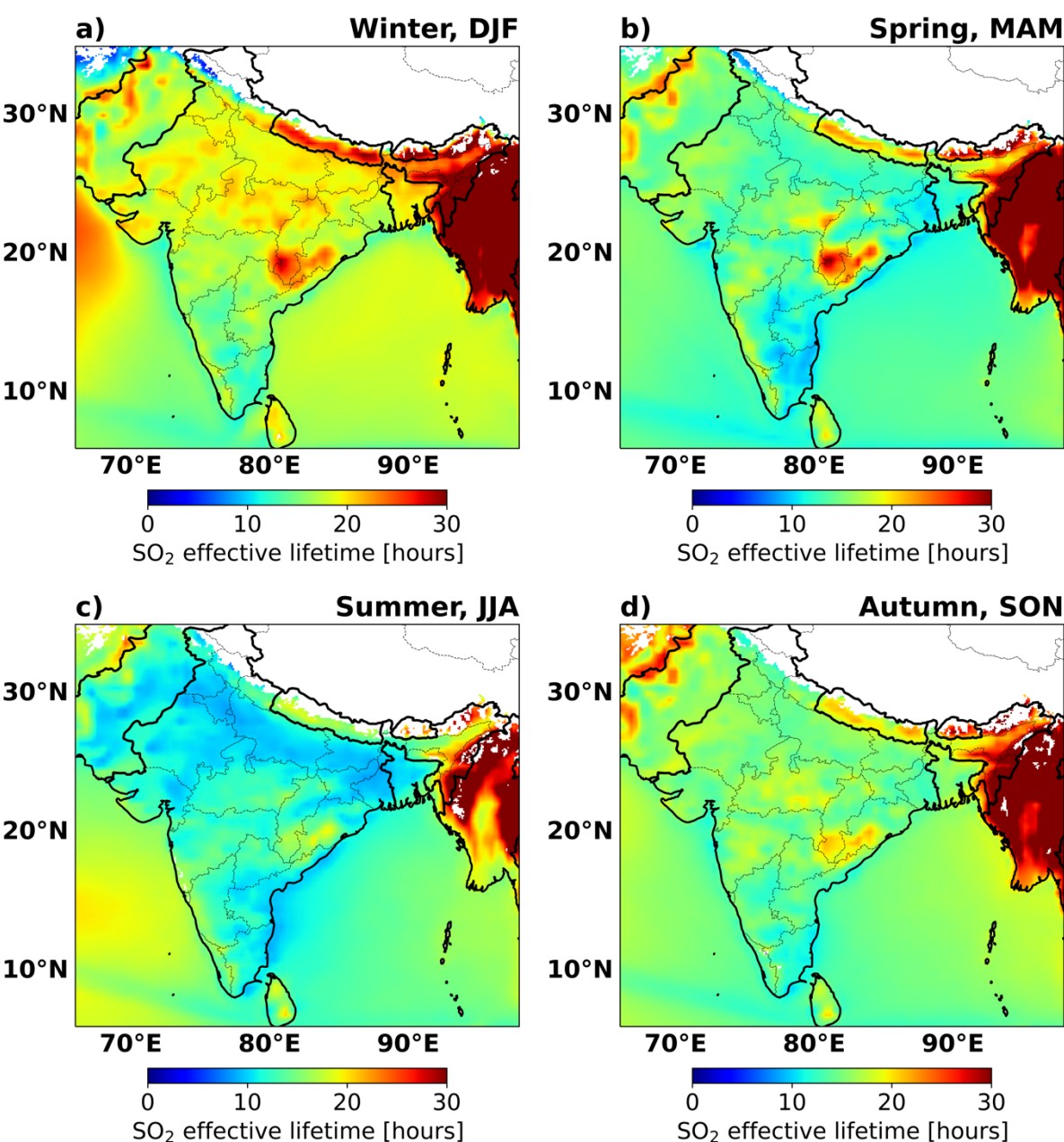

**Figure 1. SO₂ seasonal mean effective lifetime in India. Lifetime in each season is averaged for the period from December 2018 to November 2023. (a) Winter DJF: Dec-Jan-Feb; (b) Spring MAM: Mar-Apr-May; (c) Summer JJA: June-July-Aug; (d) Autumn SON: Sep-Oct-Nov. The white region represents the areas with surface heights larger than 3 km or the areas without high-quality SO₂ measurements. These regions are not discussed in this study.**

### 3.2.4 The SO₂ effective lifetime validation

The monthly mean SO₂ effective lifetime is calculated based on OH oxidation and the SO₂ dry deposition. We assume negligible influence on lifetime from SO₂ wet deposition and other chemical reactions occurring in the cloud's droplets in terms of monthly mean lifetime, especially since we use only cloud-free observations. To show this, we derive a monthly mean SO₂ lifetime ($\bar{\tau}$) from the CAMS model by considering all SO₂ producing processes and all kinds of sink according to Eq. (6),

$$\bar{\tau} = \frac{C}{E}, \tag{6}$$

with $C$ being the total $SO_2$ concentration and $E$ the total $SO_2$ emissions. We sum both the concentrations and the emissions of the model for each month covering entire India to derive a monthly mean averaged $C$ and $E$ for the whole India. Fig. 2 shows the monthly $\bar{\tau}$ in 2019-2020 and 2022-2023 based on the CAMS model. This model-intrinsic $SO_2$ lifetime of each month consistently exceeds 7 hours. The lowest lifetime is in summer, around 9.5 hours on average, while the longest lifetime is in winter, around 25.5 hours on average. The lifetime in spring and autumn is comparable, around 19 hours on average. Note that the CAMS model includes both dry and wet deposition of $SO_2$. The noticeable monthly/seasonal variation of lifetime align well with our calculations based on the OH oxidation and $SO_2$ dry deposition, indicating our calculated $SO_2$ lifetime will not change significantly even if wet deposition and other chemical reactions are considered now. At the same time, we see a large variation both spatially as in the average from month to month. Therefore, we will use the monthly-averaged local lifetime from here on.

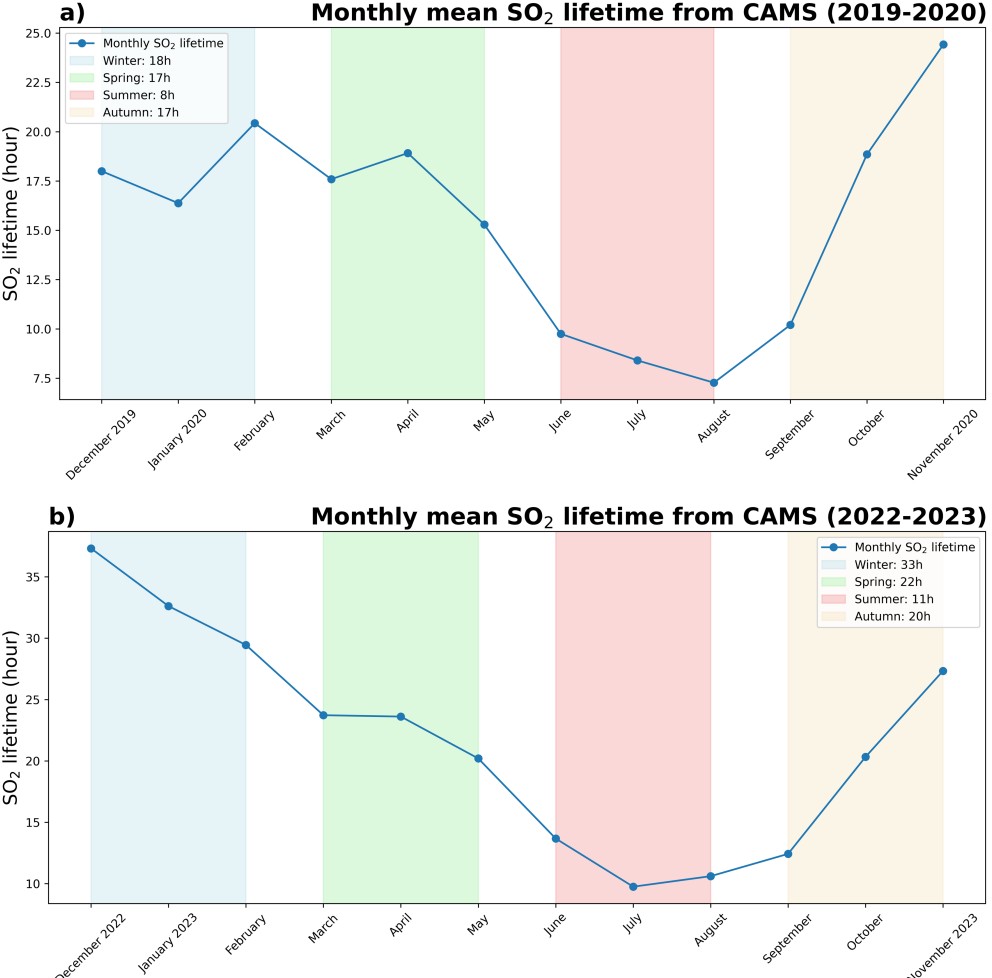

**Figure 2. Monthly averaged SO₂ lifetime in India for (a) 2019-2020 and (b) 2022-2023. The lifetime is calculated by accounting all SO₂ producing processes and all kinds of sink in the CAMS model.**

### 3.3 Emission calculation

The final emission term is the sum of the flux-divergence and sink term, and can be expressed as:

$$E = D + S = \nabla(\vec{w} \cdot V_{so_2}) + \frac{V_{so_2}}{\tau}$$

$$= \vec{w} \cdot \nabla(V_{so_2}) + V_{so_2} \cdot \nabla(\vec{w}) + \frac{V_{so_2}}{\tau}, \tag{7}$$

where $\vec{w} \cdot \nabla(V_{SO_2})$ is the flux-divergence of the SO$_2$ concentrations, $V_{SO_2} \cdot \nabla(\vec{w})$ is the wind divergence, and the last term describes the sink. The wind divergence term considers the vertical transport, which contributes to the divergence of the wind and can affect the calculated emissions. To calculate this wind divergence term , we follow the method described in Bryan (2022) to remove the wind divergence from the equation. To minimize the impact of noise on the SO$_2$ measurements, we average the divergence over each season. Emissions for each month are

then calculated by summing the monthly sink term and the divergence term of the corresponding season. The divergence calculation can be conducted on different spatial scales. Given the aimed resolution for the emissions is 0.1° × 0.1°, the divergence calculation can be conducted on a 0.1° × 0.1° regular grid cell (which corresponds to an approximate surface area of 10 km × 10 km) after integrating the measured SO$_2$ VCDs to the regular grid cells. Since the emission map resolution is limited by the pixel scale of TROPOMI, we also calculate the

divergence based on the original TROPOMI measured pixels (on an along × cross track grid, about 5.5 km× 3.5 km at nadir varying with the viewing angle) (de Foy and Schauer, 2022) and later integrate the divergence to the regular grid cells of 0.1°× 0.1°. The integration from the TROPOMI pixels to regular grid cells is based on the weight of the overlap areas. The divergence calculation on different spatial resolutions mentioned above are both conducted in this study. The final emissions are calculated from the divergence map with the best performance.

**3.4 Closed loop validation approach**

To verify both the flux-divergence method and the derived OH climatology, we have tested our method using the simulated data from CAMS forecast datasets with the known input emissions CAMS-GLOB-ANT v4.2 (Fig. 3). We use the simulated SO$_2$ VCDs within the PBL and the wind field at the mid-point of the PBL from the CAMS forecast datasets (0.4°×0.4°) from December 2019 to November 2020 to calculate the CAMS top-down SO$_2$

emissions with the flux-divergence method, in which the sink term is calculated following Section 3.2. The CAMS-GLOB-ANT v4.2 (Soulie et al., 2023), which are applied in the CAMS forecast datasets across 2019/2020, is used for comparison with the CAMS top-down SO$_2$ emissions. If they align closely, it indicates that the lifetime and flux-divergence method work well in this process.

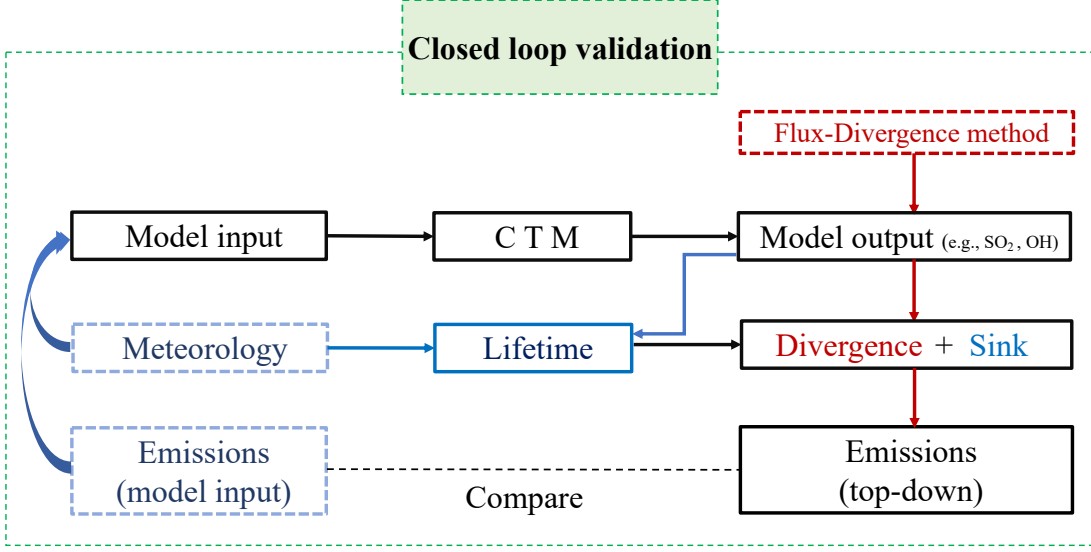

**Figure 3. Illustration of the closed loop validation.**

## 4 Improvement of the flux-divergence calculation

To verify the performance of the flux-divergence method, it is initially tested in a closed loop validation to calculate the CAMS top-down $SO_2$ emissions with a resolution of $0.4° \times 0.4°$. Figure 4a shows the model input emissions (CAMS-GLOB-ANT v4.2) and Fig. 4b shows the CAMS top-down $SO_2$ emissions derived with the original flux-divergence method (hereafter referred to as the Classic Divergence Method (CDM)). The total CAMS top-down $SO_2$ emissions for the Indian domain are 15.0 Tg year$^{-1}$, close to 13.6 Tg year$^{-1}$ calculated in the CAMS-GLOB-ANT v4.2. However, the distribution differences between the two maps are significant. The map of Fig.4a shows a more distinct emission signal at precise locations representing point-sources, whereas the emission map from Fig. 4b shows a noticeable spreading effect of point sources. This effect leads to a large difference in the emissions at the source locations. The spreading effect in the emissions derived with the CDM is a result of using the SOCFDM to approximate the continuity equation of the divergence calculation (Eq. S(1)), since it effectively involves a linear interpolation. To show this, Eq. S(1) used to calculate the divergence in grid cell $i$ along $x$ direction can be rewritten as:

$$D_{x(i)} = \frac{1}{2}\left[\frac{(\vec{F}_{x(i+1)} - \vec{F}_{x(i)})}{\Delta x} + \frac{(\vec{F}_{x(i)} - \vec{F}_{x(i-1)})}{\Delta x}\right]. \tag{8}$$

Here, $\vec{F}_{x(i)}$ denotes the flux of $SO_2$ in grid cell $i$ along the $x$ direction, and $\Delta x$ is the resolution of the grid-scale data. Then the divergence in grid cell $i$ along $x$ direction can be expressed as:

$$D_{x(i)} = \frac{1}{2}[D_{RE(i)} + D_{LE(i)}], \tag{9}$$

with $D_{RE(i)}$ and $D_{LE(i)}$ representing the divergence at the right edge and the left edge of grid cell $i$. Thus, the divergence of each grid cell is essentially a linear interpolation of the divergence at the grid cell edges. If we perceive the divergence interpolation as a divergence allocation, the linear interpolation of the divergence

essentially means that half of the divergence is allocated to the source location grid cell, while the remaining half is allocated to the grid cell adjacent to the source location grid cell, resulting in the spreading effect (Fig. S4c).

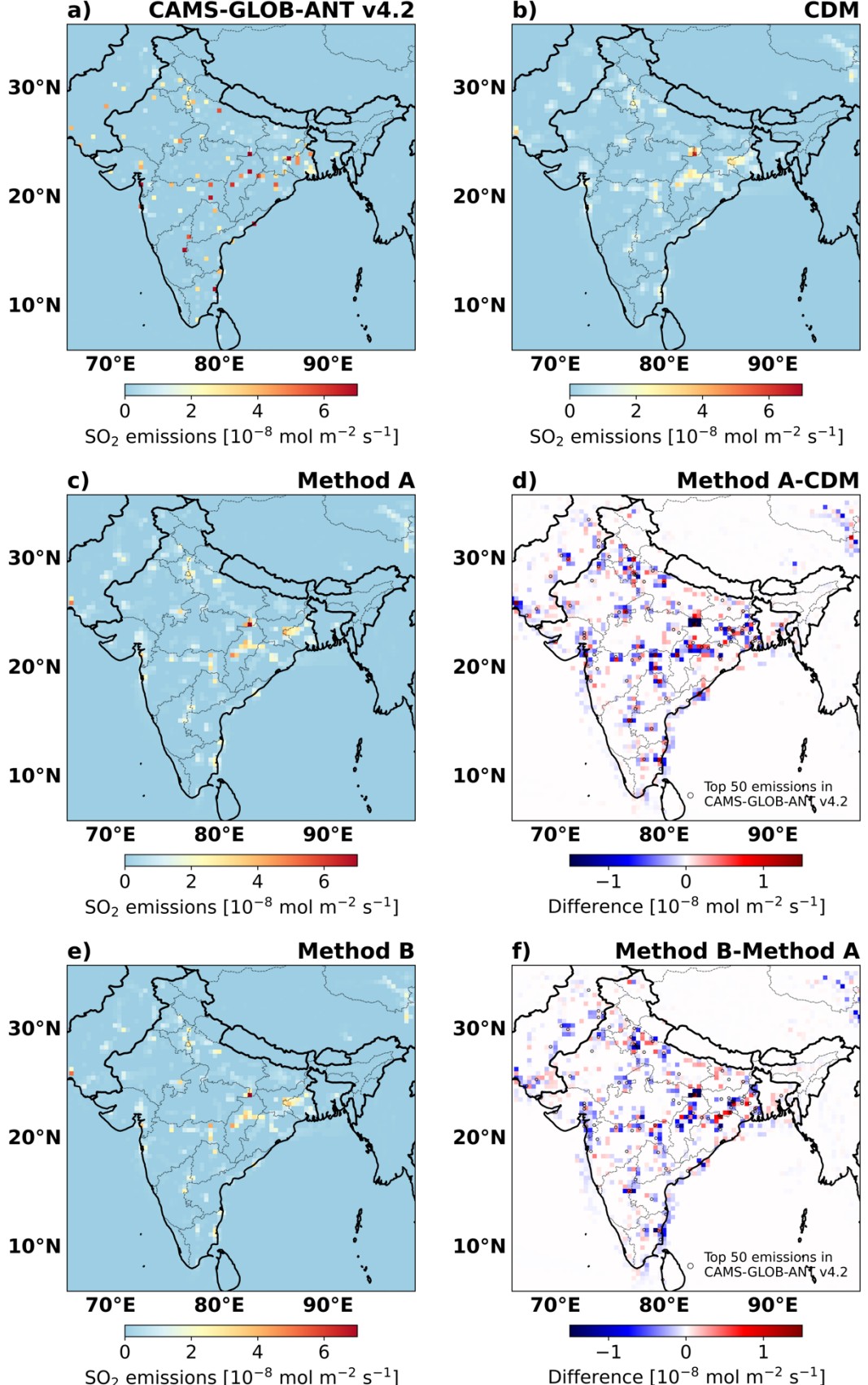

**Figure 4. The CAMS model input and CAMS top-down SO$_2$ emission distribution in the winter season (Dec-Jan-Feb) of 2019/2020. The emissions from (a) the CAMS-GLOB-ANT v4.2 inventory, and emissions derived with (b) the CDM, (c) method A, and (e) method B are shown. (d) shows the difference in emissions between the CDM and method A. (f) shows the difference in emissions between the CDM and method B. The black circles represent the locations of the top 50 emissions in the CAMS-GLOB-ANT v4.2 inventory.**

As the spreading is a result of the discrete steps in SOCFDM, the improvements mainly focus on using different divergence interpolation/allocation methods to reduce the spreading and make the emission signals "sharper" in the source locations. In the one-dimensional situation along the wind, the highest SO$_2$ concentration occurs downwind of the source (Fig. S4b). The largest SO$_2$ VCD gradient is displayed around the source especially upwind (Fig. S4a). Considering this distribution, we conduct method A, assigning all of the edge divergence to the grid cell, whose opposite edge has the larger SO$_2$ VCD gradient (see formula in Section 5 in supplementary information). Figure. 4c using method A shows that the spreading effect is reduced efficiently compared to Fig. 5b using the CDM. The most notable improvements are observed in the source locations, suggesting that method A can yield a higher-quality emission inventory. However, compared to the input emissions used in CAMS and shown in Fig. 5a, method A still shows a clear spreading effect. Although method A is very effective in a theoretical one-dimensional example, it is much less efficient in two dimensions, where the grid cells and wind direction (i.e. the plume) are usually not aligned. The highest SO$_2$ concentration downwind of the source can be dispersed across multiple grid cells in the two-dimensional situation. Therefore, the peak concentration usually occurs at the source location (See Fig. S5). Based on this, we have developed a more advanced methodology (hereafter referred to as method B), which allocates all of the edge divergence to the grid cell with the larger SO$_2$ VCD (See formula in Section 6 in supplementary information). The emission map derived with method B provide better results when compared to the CDM as shown in Fig. 5f and method A as shown in Fig. S6b. It is noteworthy that only the distribution is different between emissions derived with CDM, method A and B. The total amount of SO$_2$ emissions derived with the different methods remain the same. This is because the total divergence over the domain equals to zero and the total emission amount is solely determined by the SO$_2$ sink term. We subsequently adopt method B to calculate divergence at the resolution of 0.1°×0.1° (about 10 km×10 km) and at the original TROPOMI measured pixels (along × cross, about 5.5 km × 3.5 km at nadir), respectively. The divergence of the TROPOMI measured pixels are also gridded to 0.1°× 0.1° afterwards. From Fig. 5 we see that emissions from point sources derived from the TROPOMI measured pixels are more convergent to the point source location (less smoothing), although the background noise seems also enhanced. For each test method B shows emission maps with a higher spatial resolution than the other methods. Considering the outcome of these tests, our calculated emissions will be based on the divergence on TROPOMI measured pixels derived with method B. For emissions of an individual point source (e.g. a power plant), we will sum all emissions in the 5×5 grid cells around the point-source, because part of the spreading effect still remains in the results.

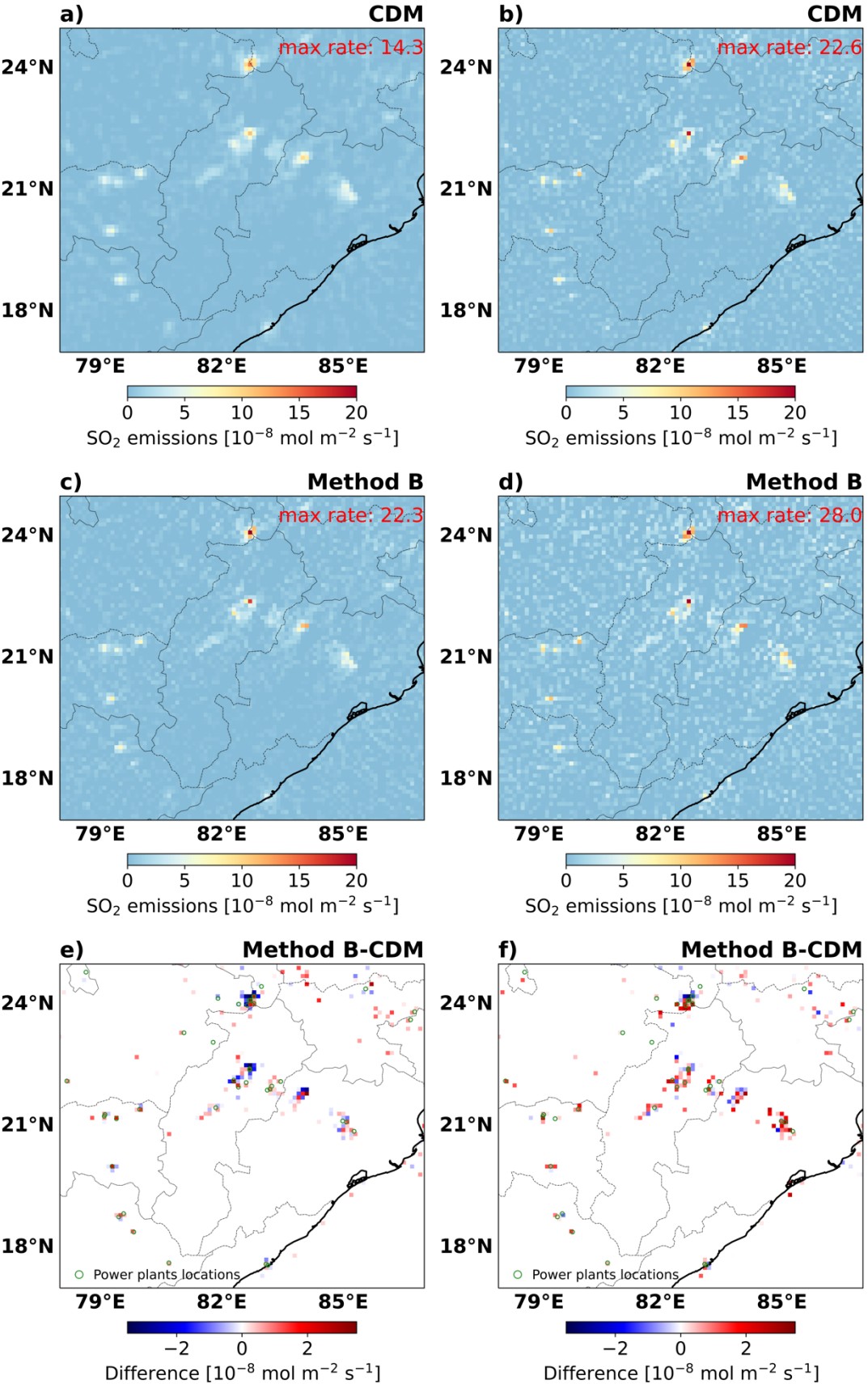

**Figure 5. The SO$_2$ emission distribution in the winter season (DJF) of 2019/2020 in a selected domain ((17°N, 25°N), (78°E, 87°E)) with large thermal power plants (a-d). The emissions (a, c) are derived from the divergence calculated directly on a 0.1° resolution using the CDM (a) and method B (c). (b, d) Emissions are derived based on the divergence calculated on the TROPOMI measured pixels using the CDM (b) and method B (d). (e, f) The difference in emissions between method B and the CDM and (method B-CDM) for the divergence calculated directly on 0.1° resolution (e) and derived on the TROPOMI measured pixels (f). The green circles represent the locations of thermal power plants with annual power generation larger than 1000MW (from Open Infrastructure Map (https://openinframap.org/stats/area/India, last access: 29 July, 2024).)**

## 5 Uncertainties assessment

As the uncertainty is mainly determined by the sink term, the SO$_2$ emissions uncertainty involves the uncertainties from the measured SO$_2$ VCDs and those associated with the SO$_2$ effective lifetime, of which the latter is primarily related to the OH concentrations and dry deposition velocity. The SO$_2$ VCDs uncertainty is mainly from the calculation of Air Mass Factors (AMFs). Here we apply an averaged AMFs uncertainty of about 30% for the individual measurement column, which is estimated from S5P/TROPOMI SO$_2$ ATBD file (https://sentinel.esa.int/documents/247904/2476257/Sentinel-5P-ATBD-SO2-TROPOMI, last access: 29 July, 2024). Considering there are 17 effective measurements on average for each month across India, the uncertainty from AMFs for monthly mean SO$_2$ VCDs is calculated to be about 7% ($\frac{30\%}{\sqrt{17}}$). The uncertainty associated with the dry deposition velocity has only a second-order effect on the SO$_2$ effective lifetime, with the uncertainty in the OH term dominating. If the dry deposition velocity increases by 100%, the effective lifetime for SO$_2$ is only reduced by 20%. Since there is a lack of validation of OH concentration due to a scarcity of measurements, we assume the differences of the simulated OH by IFS model with various chemistry scheme (IFS(CB05BASCOE), IFS(MOZART), IFS(MOCAGE)) as an estimate of the OH uncertainty, which can reach up to 50% (Huijnen et al., 2019). Changes in the OH density by ±50% generally translate to a maximum uncertainty of 60% increase or a 20% decrease in SO$_2$ effective lifetime. Consequently, the uncertainties of Indian emissions mainly involve the uncertainties from SO$_2$ VCDs and from the CAMS OH concentrations. Combining the uncertainties leads to an emission uncertainty ranging from maximum -42% to +33%. Although the measured SO$_2$ plume has no interaction with the clouds during the TROPOMI overpass, the SO$_2$ may interact with clouds before and after this time to influence the effective SO$_2$ lifetime. Therefore, we take an uncertainty of 40%, which is larger than the averaged uncertainty (35%), for SO$_2$ monthly emissions.

## 6 Results

### 6.1 Calculation of the SO$_2$ emissions and the emission detection threshold

We calculate the annual SO$_2$ emissions over India for the period December 2018 to November 2023 (5 years). The 5-year averaged annual SO$_2$ emission map in Fig. 6a effectively captures large emission hotspots. But the noise on the data hampers precise differentiation of the weakest SO$_2$ point sources. To address this, we assess the noise level on the measurement or the emission detection threshold from a selected ocean region (within (5°N-18°N) and (85°E- 90°E)), which typically contains no strong ship or other emissions. The frequency distribution of annual SO$_2$ emissions (or background noise) within the selected region approximates a normal distribution with

σ = 0.52 Gg year⁻¹ as depicted by the blue bars in Fig. S7. We define the detection threshold as four times σ (about 2.0 Gg year⁻¹ per grid cell). The emissions sources above the detection threshold are shown in Fig. 6b-d. It displays a good location alignment with the source locations detected in MSAQSO2LV4 and the known thermal power plants.

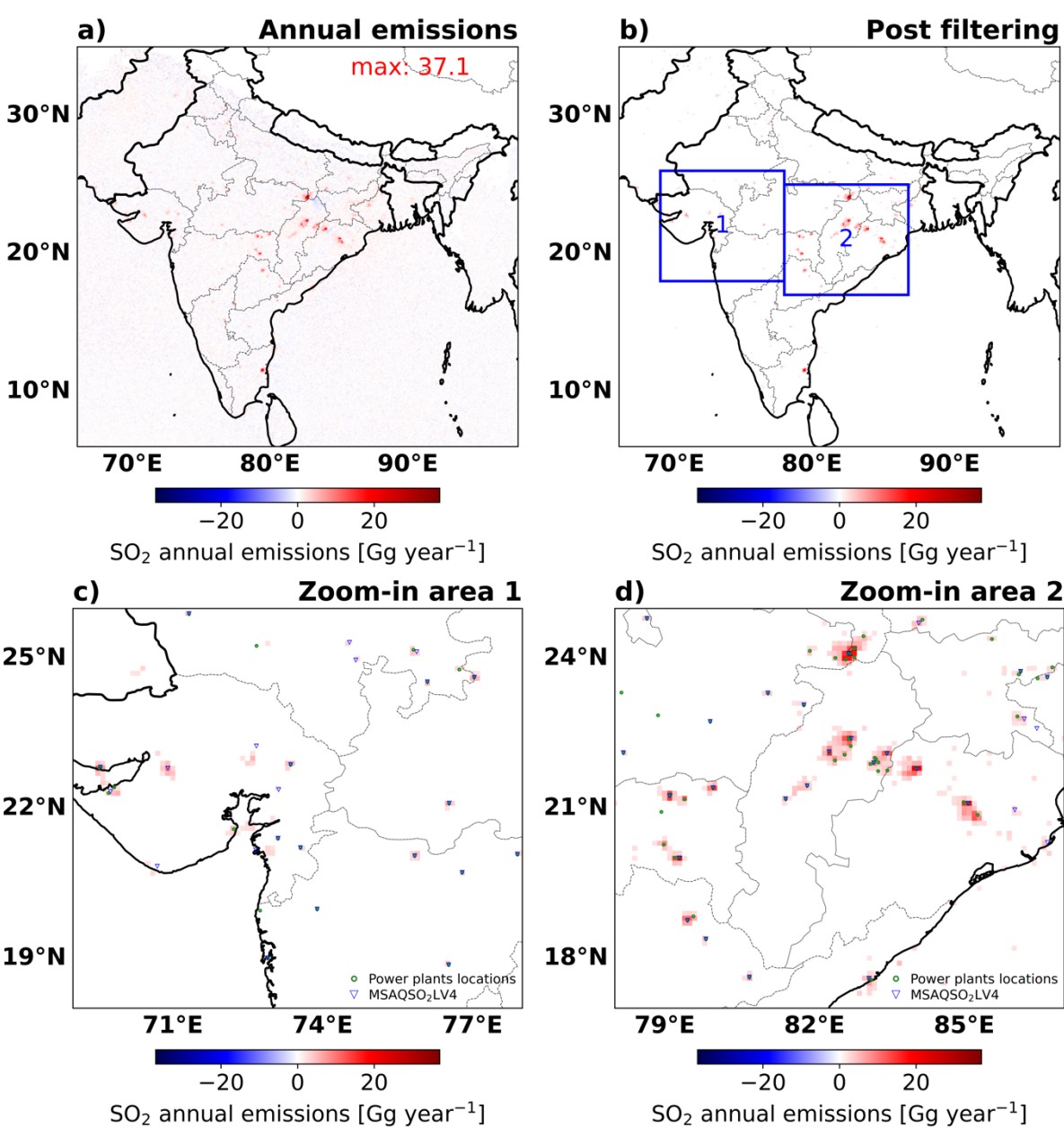

**Figure 6. (a) The SO₂ annual mean emissions averaged between December 2018 to November 2023. (b) shows the emissions above the detection threshold of 2.0 Gg year⁻¹. (c) and (d) show the emissions of zoom-in areas 1 and 2 respectively. The blue triangles represent the source locations identified by MSAQSO2L4. The green circles represent the locations of thermal power plants with annual power generation larger than 500MW from the Open Infrastructure**
**map ([https://openinframap.org/stats/area/India](https://openinframap.org/stats/area/India), last access: 29 July, 2024). The range of the color bar is scaled with the maximum value.**

The annual mean emissions for the whole of India from December 2018 to November 2023 are approximate 5.7, 4.2, 5.1 and 5.1, 5.7 Tg year⁻¹, with the 5-year averaged SO₂ emissions being 5.2 Tg year⁻¹ with an uncertainty of

$\pm 12\%$ $\left(\frac{40\%}{\sqrt{12}}\right)$. The sudden reduction in $SO_2$ emissions in 2020 corresponds to the declining trend of coal consumption in the same year (IEA, 2023) likely due to the effects of the COVID-19 pandemic on energy consumption (Levelt et al., 2022). The Indian $SO_2$ emissions show a seasonality: the emissions in winter (DJF) are on average 0.50 Tg month$^{-1}$, in spring (MAM) 0.57 Tg month$^{-1}$, in summer (JJA) 0.25 Tg month$^{-1}$ and in autumn (SON) 0.41 Tg month$^{-1}$. During the summer season more additional power capacity from hydro and wind

power is available (related to the monsoon) and less energy from coal-powerplants is needed (IEA, 2023).

## 6.2 Comparison against other Indian $SO_2$ emissions datasets

We compare our $SO_2$ emission fluxes against those taken from the global catalog MSAQSO2L4 for 92 strong $SO_2$ point-sources. The total $SO_2$ emissions of 92 point sources averaged over 5 year are 2.9 Tg year$^{-1}$, notably lower than the 5.2 Tg year$^{-1}$ in MSAQSO2L4. The scatter plot in Fig. 7 shows the annual emissions averaged over the 5

415      years study period. The strong and significant correlation (P<0.05) between the two emission datasets results in a Pearsons R value of 0.87, confirming the efficiency and accuracy of the divergence method for detection of strong point sources. To further explore the differences in these emissions terms depicted in Fig. 7a, we also calculate the emissions assuming a constant $SO_2$ lifetime of 6-hour assumed in MSAQSO2L4 by Fioletov et al. (2023). This adjustment increases our $SO_2$ emissions to 4.0 Tg year$^{-1}$, which is closer to the total emissions of the MSAQSO2L4

(Fig. 7b). But we see a noticeable smoothing effect and an overall positive bias on emissions estimated with a fixed 6-hour lifetime compared to the emissions estimated with a local, variable lifetime, especially around the source location (Fig. 7c, d and Fig. S8). This indicates that the lifetime of 6-hour is too short and the application of a non-constant $SO_2$ lifetime to constrain $SO_2$ emissions is more realistic. Consequently, we suggest that the real $SO_2$ emissions in India are lower than emissions estimated with a fixed 6-hour lifetime.

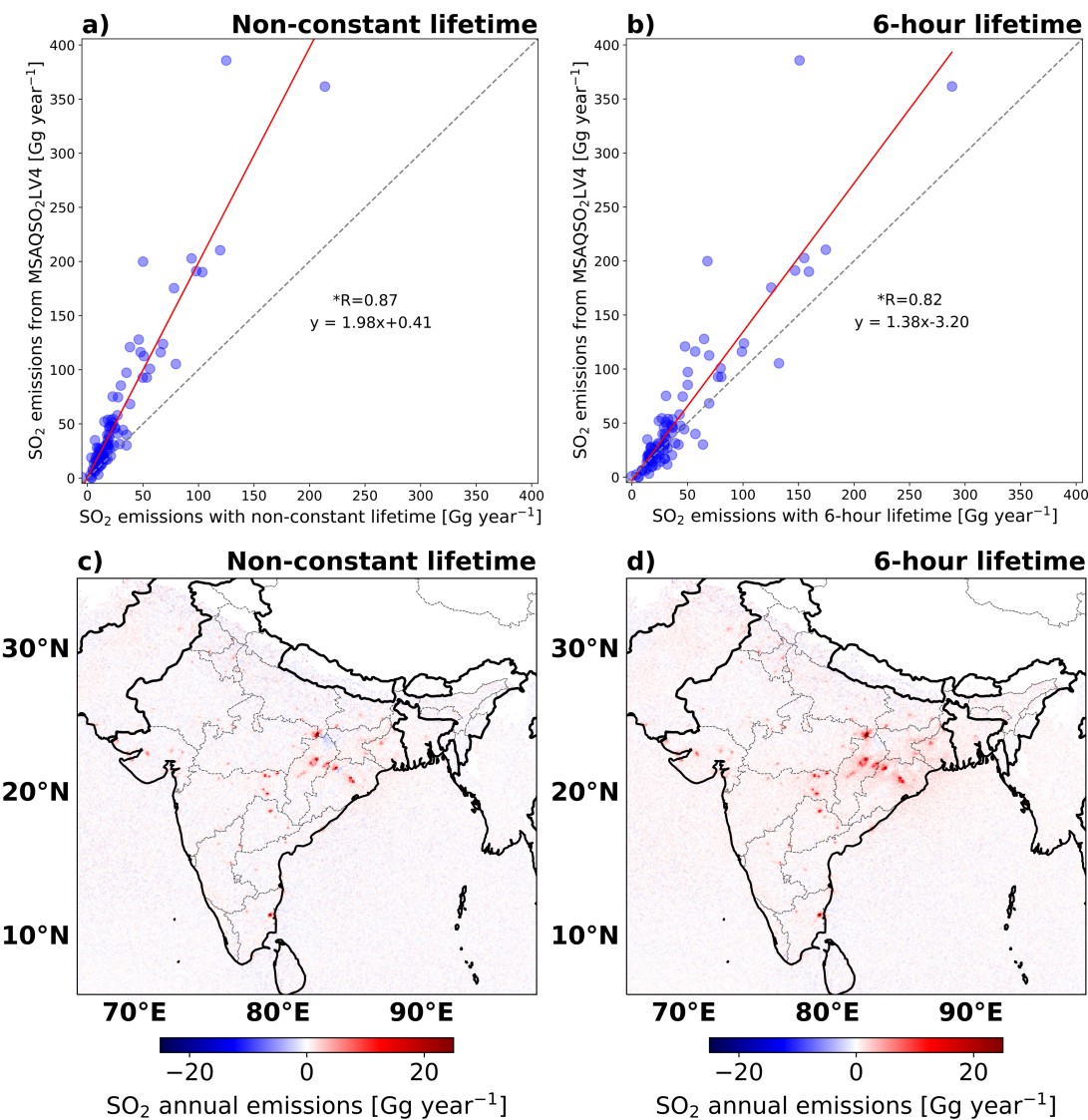

**Figure 7. (a) Comparison between SO₂ emissions in this study derived using a variable lifetime (x-axis) and the corresponding SO₂ emissions from the MSAQSO₂LV4 catalog (y-axis). (b) same as (a) but for emissions derived with a 6-hour lifetime on the x-axis. (c) SO₂ emissions derived with the non-constant lifetime. (d) same as (c) but for emissions derived with a 6-hour lifetime. The point source emissions from MSAQSO₂LV4 are averaged from 2019 to 2022. The emissions from this study are averaged from December 2018 to November 2023.**

To further compare the emissions to other inventories, we select our top 10 of highest emission sources (see locations in Fig. S9). Our top 10 sources are associated with thermal power stations, emitting in total 1.1 Tg year⁻¹, which accounts for 21% of all SO₂ emissions in India. The comparison with the global catalog MSAQSO₂LV4, and the bottom-up emission inventories, EDGAR v8 and CAMS-GLOB-ANT v5.3, are shown in Fig. 8. Generally, the emissions from our top 10 sources are lower than those reported by the other inventories. Except for Chandrapur (20.01°N, 79.29°E) and Durgaphur (23.55°N, 87.21°E), our top 10 sources are also listed in the Indian top 10 sources from MSAQSO₂LV4. The largest emitter Vindhyachal, representing 5×5 grid cells around the Vindhyachal Superpower Station (24.9°N, 82.68°E), is also the largest SO₂ emission source in CAMS-GLOB-ANT v5.3 and EDGAR v8. Neyveli (11.55°N, 79.44°E) is the largest SO₂ emitter in the MSAQSO₂L4 and is the

third largest in our inventory. Within the 5 ×5 grid cells of Neyveli, several coal power plants are situated near a lignite mine. Our comparison of the highest emitter (Neyveli) in Fig. 7a, b indicates that the emission disparities between our inventory and MSAQSO2L4 cannot be solely attributed to different lifetimes, suggesting that the choice of inversion method can also play a key role in constraining emissions.


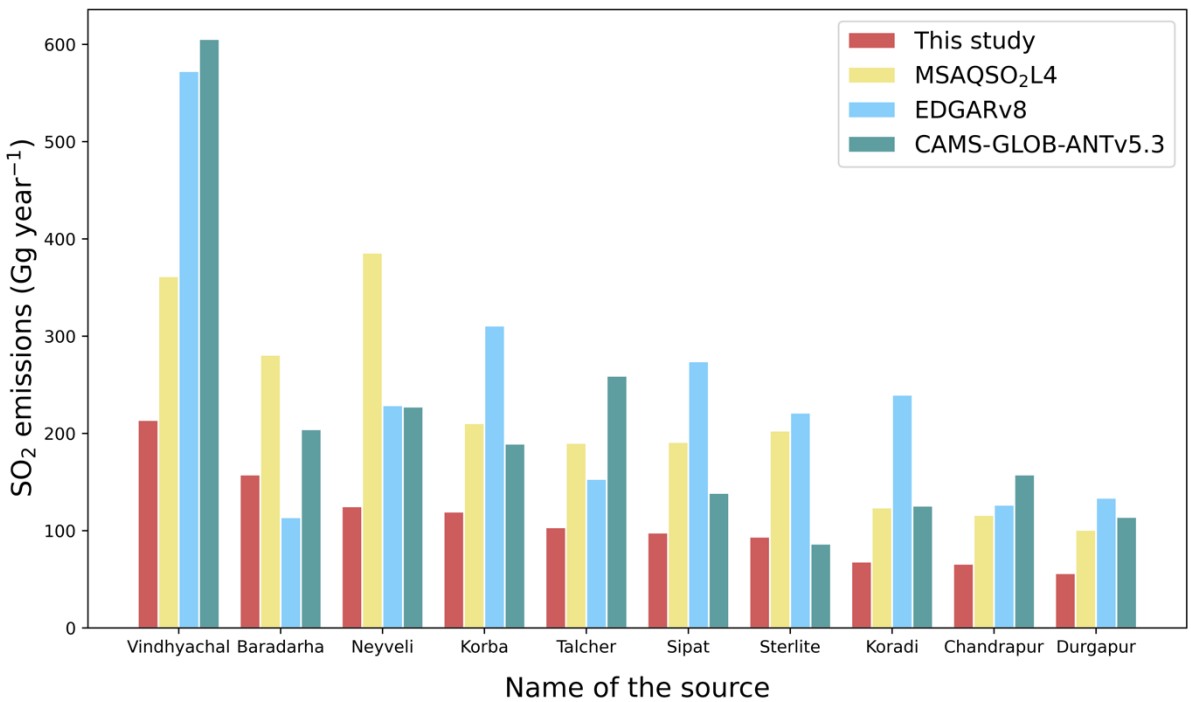

**Figure 8. A comparison of SO₂ emission estimates from our 10 largest point-sources in India using the global catalog MSAQSO₂LV4, EDGAR v8 and CAMS-GLOB-ANT v5.3 datasets. The sources are sorted by descending order of our emissions. The x-label lists the name of each source (i.e. power plant). For the inventories, the total emissions within 5×5 grid cells centered by the source location is used for comparison. Emissions from MSAQSO₂LV4 are averaged from 2019 to 2022. Emissions from EDGAR v8 are averaged from December 2018 to November 2022. Emissions from other inventories are averaged from December 2018 to November 2023.**



We calculate Indian SO$_2$ emissions to be 5.2 Tg year$^{-1}$ using the SO$_2$ local lifetime, and 12.0 Tg year$^{-1}$ using a fixed 6-hour lifetime. The country-total emission obtained with a local lifetime are about 50% lower than the reported emissions in the most used bottom-up inventories, i.e. CAMS-GLOB-ANT and EDGAR. The CAMS-GLOB-ANT v5.3 inventory estimates that India emitted 11 Tg year$^{-1}$ SO$_2$ in 2023. However, the CAMS model simulated SO$_2$ densities, driven by CAMS-GLOB-ANT v5.3, are much higher by a factor of 2 than the TROPOMI measurements (see the comparison for 2023 in Fig. 9). The emissions at the large source locations show big differences between the two maps. Many strong source signals in the west of India are shown on the CAMS map while not visible on the TROPOMI map. Considering the good data quality of the TROPOMI observations (Theys et al., 2021; De Smedt et al., 2021), we suggest that the difference in Fig. 9 is primarily due to a positive bias in model input-emissions. It is noted that the SO$_2$ lifetime in CAMS model may be overestimated and contribute to the higher simulated SO$_2$ concentration, even though the OH level, which mainly determines the SO$_2$ lifetime



under cloud-free conditions, are similar between CAMS results (Fig. S10) and the previous studies (Hewitt and Harrison, 1985; Lelieveld et al., 2004; Souri et al., 2024).

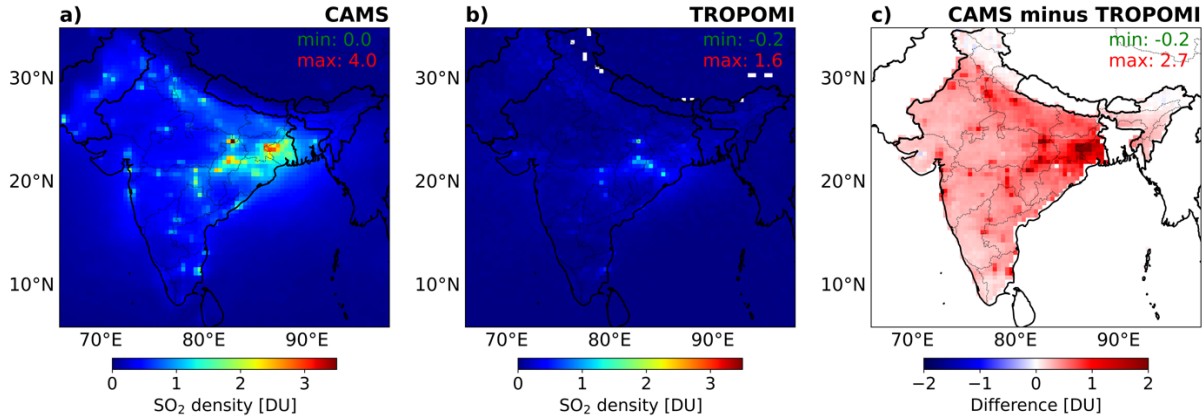

**Figure 9. Indian SO₂ vertical column densities (VCDs) averaged in 2023 from (a) CAMS global composition forecast dataset, and (b) TROPOMI Level-2 COBRA dataset (at about the overpass time of 6 UTC). We integrate the TROPOMI observations to a resolution of 0.4° × 0.4°, the same as the CAMS datasets. (c) is the difference obtained by subtracting (b) from (a). The data of the same days are used for comparison The CAMS SO₂ density with total cloud coverage larger than 30% are excluded from the averaging.**


The total SO₂ emissions in India were similar in 2019 and 2023, with lower emissions in the years in between. To explore the changes in detail, the difference in emissions between 2019 and 2023 of each point source is shown in Fig. 10. Overall, the total point source emissions are estimated to be 2.8 Tg year$^{-1}$ in 2019 and 3.0 Tg year$^{-1}$ in 2023. The point sources exhibiting the largest changes belong to our top 10 sources. The emissions of Vindhyachal,

the point source showing the largest decrease, were reduced by 17%, which is about 43 Gg year$^{-1}$. This reduction may be partially attributed to the initiation of a carbon capture project at the Vindhyachal plant started in August 2022 (PTI, 2022), which likely mitigates some of the SO₂ emissions (Wang et al., 2011; Corvisier et al., 2013; Gimeno et al., 2017). The largest increasing emitter, Baradarha, increased over 75%, which is in total 107 Gg year$^{-1}$ of SO₂ emissions.


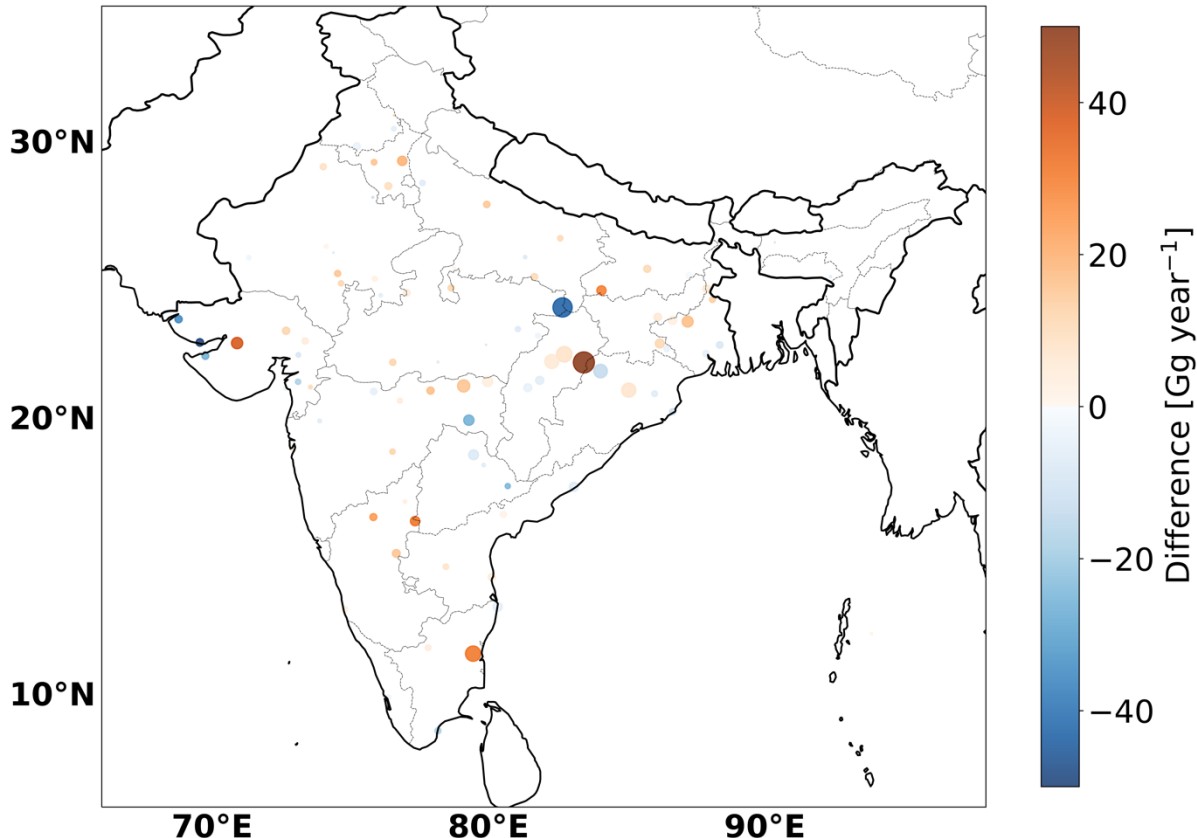

**Figure 10. Absolute changes in the derived SO₂ emissions for the most important point sources between December 2018 and November 2023. The circle size denotes the size of the emissions in the last year (December 2022 to November 2023). The circle color means changes in the last year compared to the emissions in the first year (December 2018 to November 2019).**

## 7. Discussion

The hard-to-quantify factors influencing the lifetime and emissions are discussed here. First, our grid-averaged (about 40 km × 40 km) OH climatology may does not resolve detailed chemical variation within the pollutant plumes, particularly those involving the interaction between $SO_2$ and OH. Krol et al. (2024) studied the chemistry within the $NO_x$ plumes and observed low OH concentrations near the strong $NO_x$ sources (within an average of 10 km) and enhanced OH away from the sources. This suggests that our 40 km averaged OH climatology cannot capture this OH decline and may underestimate the $SO_2$ lifetime near the large $NO_x$ sources. Furthermore, the variation of OH concentration between 10 km to 40 km is roughly limited to 10 %, see Fig. 7b from study of Krol et al. (2024). We have considered these effects in our error estimate of the lifetime. Second, we did not consider the heterogenous $SO_2$ reactions on wet aerosols. We suppose this impact on $SO_2$ lifetime can be neglected in our study. Analysis of the dominant term for conversion in the CAMS system shows aqueous phase chemistry is the dominant term related to sulphate production rather than heterogeneous reactions. We therefore assume wet aerosols are a negligible source term (which typically have low pH and slow sulphate production (Gillani et al., 1981)). Even though the atmospheric $SO_2$ in gas phase can convert to aqueous phase and be oxidized to form sulfate on aerosol wet surface or within clouds, these reactions typically occur on hazy days with high relative humidity and PM2.5 level (Ge et al., 2021). These meteorological conditions are generally not favored on days

with minimal cloud coverage, as achieving the necessary high relative humidity is difficult with ample sunlight at noon. Finally, we only calculate the lifetime for $SO_2$ in cloud-free regions, excluding the $SO_2$ wet deposition and the reactions within the clouds. This is actually the lifetime we need in our inversion since the TROPOMI observations of $SO_2$ plumes are limited to cloud-free scenes.

## 8. Conclusion

In this study, we derived the Indian $SO_2$ emissions using an improved flux-divergence method including a non-constant $SO_2$ effective lifetime for cloud-free conditions. The improved divergence method largely removes the spreading effect on emissions that is typically introduced by the discretization in calculating the divergence. The non-constant lifetime approach is more representative with respect to season and latitude as compared to adopting a fixed lifetime for the derivation of emission fluxes, especially for short-living species like $SO_2$. Based on the non-constant lifetime the improved divergence method further constrains the $SO_2$ emissions more closely to its source. The $SO_2$ effective lifetime in India for cloud-free conditions, derived from the $SO_2$ chemical lifetime and dry deposition lifetime, is calculated for each grid cell. The $SO_2$ chemical lifetime is primarily derived using an OH monthly climatology (December 2018 to November 2023) from CAMS simulations. The variability in the monthly mean $SO_2$ effective lifetime varies from 16 to 34 hours, with the longer chemical lifetime occurring in the winter season. The seasonality of the $SO_2$ chemical lifetime is driven by the OH concentration, which is largely influenced by sunlight. Significantly different chemical lifetimes were also noted across various regions within the same season. The chemical lifetime in northern India is generally larger than in the south in spring, winter, and autumn. The $SO_2$ monthly dry deposition lifetime varies from 55 to 135 hours. After accounting for the $SO_2$ dry deposition, the seasonality and regional variation of lifetime are reduced. The $SO_2$ effective lifetime is 27% lower on average compared to the chemical lifetime. The $SO_2$ monthly mean effective lifetime varies from 12 to 19 hours with the uncertainties of -20% to +60%. Our local effective lifetime calculations align with a recent study, demonstrating that the species lifetime varies spatially due to the spatial variation of the influencing factor (Krol et al., 2024).

Since the data are available in grid-scale instead of a continuous state, the divergence calculation will introduce a spreading effect to the calculated $SO_2$ divergence and emissions. To reduce the spreading effect, we have tested two divergence allocation methods on the resolution of 0.4°×0.4°, 0.1°×0.1° and TROPOMI measured pixels and concluded that assigning all flux divergence to the grid cell with the larger $SO_2$ VCD improved the results. After the implementation of the improved flux-divergence method, the smoothing of the emission map is mitigated efficiently. An emission map with more distinct emission signals has been obtained.

Implementing the improved method with a non-constant $SO_2$ lifetime, we calculated the $SO_2$ emissions for India from December 2018 to November 2023. The total annual $SO_2$ emissions we found for this period is about 5.2 Tg year$^{-1}$ with a monthly mean uncertainty of 40%. The total annual $SO_2$ emissions decreased from 2019 to 2020 due to the COVID-19 quarantine measures, then gradually increased to the same level as before COVID-19 in 2023. In contrast to the trend from MSAQSO2LV4 showing that the $SO_2$ emissions reaching its highest point in 2022, our emissions in 2022 are the same as those in 2021, and lower than the emissions in 2019 and 2023. Even though the total power generation in 2022 is higher than the previous years ([https://powermin.gov.in/en/content/power-sector-glance-all-India](https://powermin.gov.in/en/content/power-sector-glance-all-India), last access: 29 July, 2024) , the comparable emissions between 2021 and 2022 might be a

result of the growth of renewable and non-fossil fuel power generation in 2022 (https://powermin.gov.in/en/content/overview, last access: 29 July, 2024).

The 92 $SO_2$ large point sources are compared with the global catalog MSAQSO2LV4. Our total emissions of 2.9 Tg year$^{-1}$ are lower than the total emissions from MSAQSO2LV4 of 5.2 Tg year$^{-1}$. The difference is mainly because Fioletov et al. (2023) used a fixed 6-hour lifetime for calculating emissions, while our derived monthly effective

lifetimes varied from 12 to 19 hours. Using the fixed 6-hour lifetime can result in a smoothing of emission map with the divergence method and may lead to overestimation of the emissions. Our results show the $SO_2$ emissions of the 92 point sources in India are similar between 2019 and 2023. The $SO_2$ emissions at the largest point source, Vindhyachal, shows a reduction in the last years. This might be due to initiation of a carbon capture project at Vindhyachal.

With the improvement in the divergence method and locally derived variability in the lifetime, gridded $SO_2$ emissions over a large area can be estimated efficiently. This method can be applied to any region in the world to derive $SO_2$ emissions with a $0.1° \times 0.1°$ resolution based on TROPOMI observations. For those regions with more Northerly latitudes than 40°N (e.g. Northern China, Eastern Europe), the latitude and season dependent $SO_2$ lifetime with the improved divergence calculation approach has the potential to significantly improve the top-

down derivation of $SO_2$ emission estimates. This paper is considered a first step towards addressing the lifetime variability in the inversion methodology.

**Data availability**

TROPOMI COBRA $SO_2$:

https://data-portal.s5p-pal.com/products/so2cbr.html

https://distributions.aeronomie.be

Last access: 29 July, 2024

Wind field data:

https://www.ecmwf.int/en/forecasts

Last access: 29 July, 2024

CAMS global atmospheric composition forecasts:

https://ads.atmosphere.copernicus.eu/cdsapp#!/dataset/cams-global-atmospheric-composition-forecasts?tab=overview

Last access: 29 July, 2024

EDGAR v8:

https://edgar.jrc.ec.europa.eu/dataset_ap81

Last access: 29 July, 2024

CAMS-GLOB-ANT:

https://permalink.aeris-data.fr/CAMS-GLOB-ANT

Last access: 29 July, 2024

$SO_2$ global catalog MSAQSO2L4:

https://disc.gsfc.nasa.gov/datasets/MSAQSO2L4_2/summary?keywords=sulfur%20dioxide

Last access: 29 July, 2024

Indian power plants:

https://openinframap.org/stats/area/India/plants

Last access: 29 July, 2024

**Author contribution**

**Yutao Chen**: Formal analysis, writing. **Ronald J. van der A**: Physical and technical support of the divergence

method, conceptualization, paper review, and editing. **Jieying Ding**: Physical and technical support of the divergence method, conceptualization, paper review, and editing. **Henk Eskes**: Technical support of the divergence method, wind field reprocessing, paper review and editing. **Jason E. Williams**: Support of the related chemistry, paper review and editing. **Nicolas Theys**: Support of $SO_2$ COBRA datasets, paper review and editing. **Athanasios Tsikerdekis**: Support of CAMS datasets, paper review and editing. **Pieternel F. Levelt**: Paper review

and editing.

**Competing interests**

The authors declare that they have no conflict of interest.

**Financial support**

We acknowledge the funding from the China Scholarship Council (CSC).

**Acknowledgements**

We acknowledge the team of ECCAD, EDGAR, ECMWF, Copernicus Project, and all the other investigators

who have made the data used in this study and made them available online.

We acknowledge Felipe Cifuentes for inspiring the closed-loop validation approach and Lotte Bryan for developing the wind divergence removal method.

N. Theys acknowledges support from ESA S5P ATM-MPC, ESA S5P-PAL, and Belgium Prodex TRACE-S5P projects.

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
