# Peer review of "SO2 emissions derived from TROPOMI observations over India using a flux-divergence method with variable lifetimes"

_EGUsphere, 2024_

## Referee Comment (RC1)

Review of "SO2 emissions and lifetimes derived from TROPOMI observations over India using a flux-divergence method" by Chen et al.

This paper uses TROPOMI SO2 VCDs (the COBRA data product) with a variant of the flux-divergence method as pioneered by Beirle et al. (2019) to derive SO2 emissions on a 0.1 x 0.1 deg grid over India. The authors use a calculated effective lifetime assuming first order loss due to chemistry and physical removal processes that varies with season and location, with a representative value of roughly 15 hours. Their derived emissions were found to be roughly a factor of two smaller than existing bottom-up inventories and another satellite-derived SO2 emissions database.

This is a well written paper, and easy to follow. But it seems odd to me that conclusion (anthropogenic SO2 emissions in India are only half what we thought) is really just mentioned in passing. Also, there is no validation of the results… comparisons are made with databases that differ by this factor of two, but that is it. The modifications to the flux-divergence methodology as previously employed are such that they require validation in their own right. Furthermore, when one obtains results that contradict previous studies it behooves the authors to provide some rationale as to why this is, and what could be behind the bias.

While I am not convinced that the flux-divergence method is the best approach for SO2 emissions as its strength is for extended area sources (e.g., urban NOx emissions), and SO2 comes from a collection of point sources (albeit occasionally in fairly close proximity), it is worthwhile to explore its effectiveness. Retrieving emissions on a grid smears out the emissions which usually can be geolocated to within a couple km, and makes getting the total for a facility sometimes challenging since one must figure out which grid boxes need to be summed over.

Aside from the larger issue of validating this rather extra-ordinary result, my main issue, and one I see as sufficiently serious as to warrant a major rethink of this work, lay in how the lifetime is derived. The authors attempt to calculate the lifetime considering simple physical (dry deposition) and first order chemical (via SO2 + OH) loss, and these individual lifetime are combined. As an aside, why is only dry deposition included? Adding it would not impact my argument below, but it seems like wet deposition should be considered to be consistent with the approach chosen by the authors. Dry depsotion lifetime were quite long, >60 hours, and thus do not have a huge impact on the combined lifetime. Chemical lifetime were calculated using an OH field ($\tau = 1/[k\,[OH])$ from the ECMWF CAMS forecast model, and are roughly 20 hours, varying in space and time. The combined lifetime is something like 15 hours.

My concerns regarding this lifetime issue are laid out here:

1. Using the OH field from ECMWF CAMS is not appropriate as the spatial resolution is 0.4 x 0.4 degrees (~40 km), and thus represent an average OH value, with half or more of the averaging area (on average; including the upwind portion) representing background values and chemistry. What is relevant is OH in the plume where the bulk of the SO2 is, and OH at the plume core, plume edge, and background will all be much different. Using different models with comparable resolution to look at differences does not help in this regard. If one wants to use model OH then something like a plume-following chemical box model is the most appropriate choice.

2. In previous applications of the flux-divergence method, the lifetime was derived using the data itself (including and especially in the original formulation by Beirle et al. (2019)… and this seems

like the obvious method to employ here. Complications such as non-linear chemistry will then be accounted for. There is sufficient signal to tease out some spatial and seasonal differences from the TROPOMI data itself. At the very least the authors should have validated their calculated lifetimes using the data itself!

3. A simple analysis of the plumes themselves in the satellite data suggests that the effective lifetime is shorter than 15 hours. The authors do not ever show the actual TROPOMI $SO_2$ data in their paper (there is one panel in the supplement), which seems strange considering it is the basis for the emissions calculations. Shown below in Figure 1 is an OMI $SO_2$ VCD averaged over 2014-2017 (this is all I had handy; TROPOMI will look similar, but hot spots will appear sharper due to its higher spatial resolution). The dots are the larger $SO_2$ sources. These are unpublished, diagnostic figures from the same EMG (exponentially modified Gaussian) method as published in Fioletov et al. (2016, 2018), McLinden et al. (2016) and later papers by that group. The left panel is the mean VCD (minus a slowly varying background bias, an artefact of the method, as discussed in the references above). The right is the reconstruction of the satellite data assuming a 6 hour effective lifetime.

[Figure]

*Figure 1: SO2 VCDs over India. Left: Observed OMI VCDs, with a large scale bias removed. Right: Reconstructed (and accounting for the resolution of the satellite) using the Fioletov et al. method assuming a six hour lifetime.*

The maps above, with the hotspots so closely linked to their emissions location, suggest a 6 hour lifetime is about right. Increasing this to 15 hours would smear the $SO_2$ out to a point where it would look like a smoothed version of the observations.

4. For EMG methods similar to that employed in my point 3, de Foy et al. (2015) argued that there is a dispersion component to the effective lifetime, and it reduces the overall lifetime relative to

the chemical+physical lifetime.  In my understanding of the flux divergence method, diffusion is not accounted for and so it seems reasonable that the effective lifetime appropriate for this method may also have a dispersive component.  Further, in the case of an isolated point source, as it the case for many here, the two methods are virtually the same with the main difference being that the EMG method of Fioletov et al. assumes a plume shape, and the flux divergence method does not constrain the plume, but derives emissions on a grid.  If this case, the physical interpretation and value of the effective lifetime should be very similar, and I would argue that my Figure 1 above is the most compelling argument for a shorter lifetime.  See Vindhyachal in particular (the most northerly isolated hotspot).  If the physical interpretation of the lifetime is not the same, then one would need to look at how the different methods perform when emissions are well know, such as from CEMS or other direct measurements.  The EMG method of Fioletov et al has been extensively validated against such measurements in the US, Europe, and for volcanic sources as well.

5.  Application to India is more difficult since there is little reference data to validate against.  In such cases it is useful to apply the method to other locations where such data is available. For example, in the US, there is CEMS data to compare against.  The authors even mention it can applied anywhere.  If one wants their results to be received with confidence, this is an essential step.

Additional comments:

Presumably there is nothing that unusual about Indian OH levels (the lifetime as derived here, to first order only depends on k and [OH]), and thus had the authors preformed a global analysis they would have found similar results (ie, ~a factor of two smaller emissions as compared to current, best emissions values) everywhere.  Even just for India this has large implications, but for the globe this would lead us to rethink the SO2 budget.

Line 382:  "But we see a noticeable smoothing effect and an overall positive bias on emissions estimated with a fixed 6-hour lifetime compared to the emissions estimated with a local, variable lifetime, especially around the source location"

But given the colour-scale, and comparing Figure S8 with Figure 5, this is exactly what one would expect when you ~double the emissions… there should be more red because all values are larger for the 6 hour lifetime, and thus it would appear to be smoother.  The only way this line of reasoning could be reliably used is to compare an isolated source, and then normalize them their peak value.   Eyeballing Vindhyachal, the large isolated point source in my Figure 2, below, the effect, even with the 2x emissions, appears marginal.  Bigger picture, this type of analysis is indirect at best… again, why not use the actual data to deduce the lifetime?

[Figure]

*Figure 2:  left Figure 5d (longer, variable lifetime); right: Figure S8d, 6 hour lifetime.*

In closing, I would like to say that this study has merit.  However, the approach used to derive the lifetime is quite concerning, and it has very large implications for the SO2 budget.  Therefore as it currently stands I feel this paper should be rejected until the points above are addressed in a comprehensive manner.

References:

De Foy (2015) https://www.sciencedirect.com/science/article/pii/S1352231015301291

Fioletov et al (2016)  www.atmos-chem-phys.net/16/11497/2016/

Fioletov et al (2018) https://doi.org/10.5194/acp-17-12597-2017

McLinden et al. (2016) https://www.nature.com/articles/ngeo2724

---

## Referee Comment (RC2)

This paper uses a flux-divergence method to estimate monthly SO2 emissions in India based on satellite observations. The paper is well organized but I have a little concern about the calculation of SO2 lifetimes, which is crucial to the final emission estimation. The authors declaimed that the satellite overpasses at noon time and they used SO2 under cloud-free conditions, so they considered only the gas phase loss of SO2. It may be true that SO2 is mainly removed through the reaction with OH as the satellite overpasses, but SO2 lifetime is as long as tens of hours, during this time, SO2 is removed mostly by the liquid phase or the heterogeneous reactions, both of which is faster and more efficient than the gas phase reaction. So I'm not sure if it is appropriate to consider only gas phase loss of SO2 in the SO2 lifetime estimation. The SO2 emission estimated here is much less (even 50%) than other datasets, does this indicate that the SO2 lifetime calculated in this study is too long?

And a minor comment, line 213: 'the SO2 monthly dry deposition lifetime within the PBL height is calculated by dividing the PBL height by 0.4 cm s-1', shouldn't it be '... dividing half the PBL height by 0.4 cm s-1'?

Concerning these, I'd like to recommend a major revision about the SO2 lifetime estimation.

---

## Author Comment (AC2)

**Response to reviews**

We thank all the reviewers for spending time on the paper improvement. Below, we provide a point-by-point response to the comments and we list the related changes made to the manuscript. The original review text is indicated in italic blue font and the response in regular black font. New text appearing in the revised paper is indicated by a red color.

**Reviewer Mclinden**,**

This is a well written paper, and easy to follow. But it seems odd to me that conclusion (anthropogenic  $SO_2$  emissions in India are only half what we thought) is really just mentioned in passing. Also, there is no validation of the results... comparisons are made with databases that differ by this factor of two, but that is it. The modifications to the flux-divergence methodology as previously employed are such that they require validation in their own right. Furthermore, when one obtains results that contradict previous studies it behooves the authors to provide some rationale as to why this is, and what could be behind the bias.

Information on the Indian SO2 emissions is sparse, which limits the validation of the emissions obtained. We also do not have access to the in-situ measurements of India SO2 surface concentrations for recent years. Therefore, we limited our comparisons to bottom-up inventories and the satellite-based emission estimates of Fioletov et al. (2023). Since our resulting emissions are about 50% lower than those of previous results, i.e. EDGAR v8, CAMS-GLOB-ANT v5.3 and estimation of Fioletov et al. (2023), we provide additional evidence to support our results in the revised manuscript.

In the bottom-up emission inventories, i.e. the recent versions of EDGAR and CAMS-GLOB-ANT, the country-total Indian SO2 anthropogenic emissions are in the range of 10-14 Tg/year. According to the CAMS-GLOB-ANT v5.3 inventory India emitted approximately 11 Tg/year in 2023. However, we found that the CAMS model-simulated SO2 columns, which are driven by the CAMS-GLOB-ANT v5.3, are much higher by a factor of 2 than TROPOMI SO2 measurements in recent years (see the comparison in 2023 in Fig. 9 and below). Considering the good data quality of TROPOMI observations (Theys et al., 2021; De Smedt et al., 2021) and assuming the CAMS model has a good performance, we attribute the higher simulation results for a large part to the overestimation of the model input emissions.

It is important to note that if we use a constant SO2 lifetime of 6 hours, this will result in an SO2 emission total of approximately 12.0 Tg/year for India, aligning well with the estimated emissions of bottom-up inventories. This information is now added in the revised manuscript from line 445. However, instead of taking a fixed lifetime, we have made an extra effort to derive the local SO2 lifetime. We notice that the factors determining the SO2 lifetime can vary dramatically both spatially and temporally, leading to a significant variability of the lifetime. Previous studies indicated a wide range of SO2 lifetimes (about 0.6-2.6 days in Lee et al., 2011 and references therein) rather than a fixed value. Therefore, we assume that a constant lifetime is not the best choice to calculate emissions for the whole India.

**We add the discussion started from line 445 in the revised manuscript:**

We calculate Indian SO2 emissions to be 5.2 Tg year-1 using the SO2 local lifetime, and 12.0 Tg year-1 using a fixed 6-hour lifetime. The country-total emission obtained with a local lifetime are about 50% lower than the reported emissions in the most used bottom-up inventories, i.e. CAMS-GLOB-ANT and EDGAR. The CAMS-GLOB-ANT v5.3 inventory estimates that India emitted 11 Tg year-1 SO2 in 2023. However, the CAMS model simulated SO2 densities, driven by CAMS-GLOB-ANT v5.3, are much higher by a factor of 2 than the TROPOMI measurements (see the comparison for 2023 in Fig. 9). Considering the good data quality of the TROPOMI observations (Theys et al., 2021; De Smedt et al., 2021) and assuming the CAMS model has a good performance, we attribute the higher simulation results mainly to a positive bias in the emissions that were input to the model.

Figure 9. Indian SO2 vertical column densities (VCDs) averaged in 2023 from (a) CAMS global composition forecast dataset (at about the overpass time of 6 UTC), and (b) TROPOMI Level-2 COBRA dataset. We integrate the TROPOMI observations to a resolution of  $0.4^{\circ} \times 0.4^{\circ}$ , the same as the CAMS datasets. (c) is the difference obtained by subtracting (b) from (a). The data of the

**same days are used for comparison. The CAMS SO2 density with total cloud coverage larger than 30% are excluded from the averaging.**

While I am not convinced that the flux-divergence method is the best approach for  $SO_2$  emissions as its strength is for extended area sources (e.g., urban NOx emissions), and  $SO_2$  comes from a collection of point sources (albeit occasionally in fairly close proximity), it is worthwhile to explore its effectiveness. Retrieving emissions on a grid smears out the emissions which usually can be geolocated to within a couple km, and makes getting the total for a facility sometimes challenging since one must figure out which grid boxes need to be summed over.

The divergence method is fast and no model or other prior knowledge is needed for the emission derivation. The method does not need clearly isolated plumes, and it works also for areas with a mix of emissions (highways, towns, industry, power plants close together). The daily estimates from the divergence method and a longer time period of observations are needed. Up to now, the flux-divergence method has been proved to be effective for point source of NOx (Beirle et al., 2019), CH4 (Liu et al., 2021) and CO2 (Hakkarainen et al., 2022). In our study we explore the feasibility for SO2 emissions and have discussed the spreading effect from the divergence method. One of our major efforts is to reduce the spreading caused by the divergence calculation. |With this improvement, the determination of the location of the SO2 point sources improved.

The authors attempt to calculate the lifetime considering simple physical (dry deposition) and first order chemical (via  $SO_2 + OH$ ) loss, and these individual lifetime are combined. As an aside, why is only dry deposition included? Adding it would not impact my argument below, but it seems like wet deposition should be considered to be consistent with the approach chosen by the authors. Dry deposition lifetime were quite long, >60 hours, and thus do not have a huge impact on the combined lifetime. Chemical lifetime were calculated using an OH field ( $\tau = 1/[k [OH])$  from the ECMWF CAMS forecast model, and are roughly 20 hours, varying in space and time. The combined lifetime is something like 15 hours.

In our study, we calculate the monthly mean SO2 effective lifetime under the PBL height, as the major amount of SO2 is concentrated inside the PBL. As a major oxidant in the troposphere (Crutzen, 1973; Logan et al., 1981), the OH concentration is high in India (Lelieveld et al., 2004; Lelieveld et al., 2016).Therefore, we assume the OH oxidation plays the most important role in determining the SO2 lifetime. The SO2 dry deposition, which occurs at the same time, has indeed a secondary influence on the lifetime (Chin et al., 1996). The SO2 wet deposition and other chemical reactions occurring in the cloud's droplets are less important (Smith and Jeffrey, 1975; Chin et al., 1996; Qu et al., 2019) compared to the OH oxidation and SO2 dry deposition in terms of the monthly mean lifetime, especially since we use only cloud-free observations. In this case, we suggest that the SO2 monthly mean lifetime will not change significantly even we involve the wet deposition and other chemical reactions. (See the answer of Point 2 below for more information.)

**My concerns regarding this lifetime issue are laid out here:**

1.Using the OH field from ECMWF CAMS is not appropriate as the spatial resolution is 0.4 x 0.4 degrees (~40 km), and thus represent an average OH value, with half or more of the averaging area (on average; including the upwind portion) representing background values and chemistry. What is relevant is OH in the plume where the bulk of the  $SO_2$  is, and OH at the plume core, plume edge, and background will all be much different. Using different models with comparable resolution to look at differences does not help in this regard. If one wants to use model OH then something like a plume-following chemical box model is the most appropriate choice.

Just as the reviewer mentioned, the variation of the lifetime influencing factors should be considered. That is why we calculate the lifetime on each grid-cell. Solving the varied lifetime both in and out of the plume would be the way forward in the future. In this paper, we have made a first step by using CAMS output, which is indeed coarse but is already showing that concentrations and lifetimes can vary spatially and temporally and differ from simplified constant lifetime assumptions. We think it is a good start to derive the non-constant local lifetime for emission estimation.

Additionally, the OH changing within the plume is not fully ignored in our method although the resolution of the model grid is relatively coarse. To support this statement, we calculate the distance that is possible for  $SO_2$  transportation according to the simple exponential decay function Eq. (1) and distance calculation function Eq. (2)

$$C = C_0 e^{-\frac{t}{\tau}},\tag{1}$$

$$S = W t \tag{2}$$

In the given equations, *C* represents the SO2 concentration after it has decayed over a time *t*, given a lifetime  $\tau$  and an initial concentration *C*0. *S* denotes the distance travelled by the plume

over the time t at wind speed W. If we assume the initial concentration  $C_0$  is 1 and C is 10%, the possible distance at different wind speeds and lifetime conditions are listed below.

|        | $\tau = 6h$ | $\tau = 15h$ |
|--------|-------------|--------------|
| W=1m/s | S=49.8 km   | S=124.3 km   |
| W=2m/s | S=99.6 km   | S=248.7 km   |
| W=3m/s | S=149.4 km  | S=373.0 km   |
| W=4m/s | S=199.2 km  | S=497.4 km   |

Based on the table, we know that the  $SO_2$  plume can cover more than two model grids with a resolution of 0.4° (about 40 km) even under very low wind conditions (2 m/s) and a short  $SO_2$  lifetime (6 hours). Therefore, the OH change along the plume is considered on the resolution of our grid.

2.In previous applications of the flux-divergence method, the lifetime was derived using the data itself (including and especially in the original formulation by Beirle et al. (2019) and this seems like the obvious method to employ here. Complications such as non-linear chemistry will then be accounted for. There is sufficient signal to tease out some spatial and seasonal differences from the TROPOMI data itself. At the very least the authors should have validated their calculated lifetimes using the data itself!

In the background, without local emissions, we expect that the negative values from the flux divergence are cancelled out by the positive lifetime term. Therefore, the lifetime is optimised when this cancellation occurs in the divergence data. Figure S8 with fixed 6h lifetime seems to suggest a higher positive background as compared to Figure 6 with local, which indicates that the local lifetime is more realistic.

We further checked the averaged lifetime by deriving a new monthly mean SO2 lifetime ( $\bar{\tau}$ ) from the CAMS model by considering all SO2 consuming processes and all kinds of sink according to Eq. (3),

$$\bar{\tau} = \frac{C}{E},\tag{3}$$

With *C* being the SO2 concentration and *E* the SO2 emission rate. We sum *M* and *E* for each month covering entire India to derive a monthly mean  $\bar{\tau}$  averaged for the whole India. Fig. 2

shows the monthly  $\bar{\tau}$  in 2019-2020 and 2022-2023 based on the CAMS model. The lowest lifetime is in summer, around 9.5 hours on average, while the longest lifetime is in winter, around 25.5 hours on average. The lifetime in spring and autumn is comparable, around 19 hours on average. The noticeable monthly/seasonal variation of lifetime align well with our calculations based on the OH oxidation and SO2 dry deposition. This indicates that our calculated SO2 lifetime will not change significantly even if wet deposition and other chemical reactions are considered now. Additionally, it is important to note that this model-intrinsic SO2 lifetime of each month consistently exceeds 7 hours. It is evidence that a fixed lifetime is not the best choice for the emission estimation for the whole India.

In addition, we set a closed loop validation to validate the calculated  $SO_2$  lifetime from CAMS model in Sec. 3.4 in the paper. This validation shows the emissions derived from the  $SO_2$  lifetimes are comparable with the model input emissions, indicating that the lifetime we derive are in line with the model. Note that we account for the high uncertainty of the lifetime in the estimated error of the final emissions.

We add section 3.2.4 in the revised manuscript:

The monthly mean SO2 effective lifetime is calculated based on OH oxidation and the SO2 dry deposition. We assume negligible influence on lifetime from SO2 wet deposition and other chemical reactions occurring in the cloud's droplets in terms of monthly mean lifetime, especially since we use only cloud-free observations. To show this, we derive a new monthly mean SO2 lifetime ( $\bar{\tau}$ ) from the CAMS model by considering all SO2 producing processes and all kinds of sink according to Eq. (6),

$$\bar{\tau} = \frac{C}{E},\tag{6}$$

with *C* being the total SO2 concentration and *E* the total SO2 emissions. We sum both the concentrations and the emissions of the model for each month covering the entire India to derive a monthly mean averaged *C* and *E* for the whole India. Fig. 2 shows the monthly  $\bar{\tau}$  in 2019-2020 and 2022-2023 based on the CAMS model. This model-intrinsic SO2 lifetime of each month consistently exceeds 7 hours. The lowest lifetime is in summer, around 9.5 hours on average, while the longest lifetime is in winter, around 25.5 hours on average. The lifetime in spring and autumn is comparable, around 19 hours on average. Note that the CAMS model

includes both dry and wet deposition of SO2. The noticeable monthly/seasonal variation of lifetime align well with our calculations based on the OH oxidation and SO2 dry deposition, indicating our calculated SO2 lifetime will not change significantly even if wet deposition and other chemical reactions are considered. At the same time, we see a large variation both spatially as in the average from month to month. Therefore, we will use the monthly-averaged local lifetime from here on.

---

## Author Response (AR1)

**Response to reviews**

We thank all the reviewers for spending time on the paper improvement. Below, we provide a point-by-point response to the comments and we list the related changes made to the manuscript. The original review text is indicated in italic blue font and the response in regular black font. New text appearing in the revised paper is indicated by a red color.

**Reviewer Mclinden,**

*This is a well written paper, and easy to follow. But it seems odd to me that conclusion (anthropogenic SO₂ emissions in India are only half what we thought) is really just mentioned in passing. Also, there is no validation of the results... comparisons are made with databases that differ by this factor of two, but that is it. The modifications to the flux-divergence methodology as previously employed are such that they require validation in their own right. Furthermore, when one obtains results that contradict previous studies it behooves the authors to provide some rationale as to why this is, and what could be behind the bias.*

Information on the Indian $SO_2$ emissions is sparse, which limits the validation of the emissions obtained. We also do not have access to the in-situ measurements of India $SO_2$ surface concentrations for recent years. Therefore, we limited our comparisons to bottom-up inventories and the satellite-based emission estimates of Fioletov et al. (2023). Since our resulting emissions are about 50% lower than those of previous results, i.e. EDGAR v8, CAMS-GLOB-ANT v5.3 and estimation of Fioletov et al. (2023), we provide additional evidence to support our results in the revised manuscript.

In the bottom-up emission inventories, i.e. the recent versions of EDGAR and CAMS-GLOB-ANT, the country-total Indian $SO_2$ anthropogenic emissions are in the range of 10-14 Tg/year. According to the CAMS-GLOB-ANT v5.3 inventory India emitted approximately 11 Tg/year in 2023. However, we found that the CAMS model-simulated $SO_2$ columns, which are driven by the CAMS-GLOB-ANT v5.3, are much higher by a factor of 2 than TROPOMI $SO_2$ measurements in recent years (see the comparison in 2023 in Fig. 9 and below). Considering the good data quality of TROPOMI observations (Theys et al., 2021; De Smedt et al., 2021) and assuming the CAMS model has a good performance, we attribute the higher simulation results for a large part to the overestimation of the model input emissions.

It is important to note that if we use a constant $SO_2$ lifetime of 6 hours, this will result in an $SO_2$ emission total of approximately 12.0 Tg/year for India, aligning well with the estimated emissions of bottom-up inventories. This information is now added in the revised manuscript from line 445. However, instead of taking a fixed lifetime, we have made an extra effort to derive the local $SO_2$ lifetime. We notice that the factors determining the $SO_2$ lifetime can vary dramatically both spatially and temporally, leading to a significant variability of the lifetime. Previous studies indicated a wide range of $SO_2$ lifetimes (about 0.6-2.6 days in Lee et al., 2011 and references therein) rather than a fixed value. Therefore, we assume that a constant lifetime is not the best choice to calculate emissions for the whole India.

We add the discussion started from line 445 in the revised manuscript:

We calculate Indian $SO_2$ emissions to be 5.2 Tg year$^{-1}$ using the $SO_2$ local lifetime, and 12.0 Tg year$^{-1}$ using a fixed 6-hour lifetime. The country-total emission obtained with a local lifetime are about 50% lower than the reported emissions in the most used bottom-up inventories, i.e. CAMS-GLOB-ANT and EDGAR. The CAMS-GLOB-ANT v5.3 inventory estimates that India emitted 11 Tg year$^{-1}$ $SO_2$ in 2023. However, the CAMS model simulated $SO_2$ densities, driven by CAMS-GLOB-ANT v5.3, are much higher by a factor of 2 than the TROPOMI measurements (see the comparison for 2023 in Fig. 9). Considering the good data quality of the TROPOMI observations (Theys et al., 2021; De Smedt et al., 2021) and assuming the CAMS model has a good performance, we attribute the higher simulation results mainly to a positive bias in the emissions that were input to the model.

[Figure]

**Figure 9. Indian $SO_2$ vertical column densities (VCDs) averaged in 2023 from (a) CAMS global composition forecast dataset (at about the overpass time of 6 UTC), and (b) TROPOMI Level-2 COBRA dataset. We integrate the TROPOMI observations to a resolution of 0.4° × 0.4°, the same as the CAMS datasets. (c) is the difference obtained by subtracting (b) from (a). The data of the**

**same days are used for comparison. The CAMS SO₂ density with total cloud coverage larger than 30% are excluded from the averaging.**

*While I am not convinced that the flux-divergence method is the best approach for SO₂ emissions as its strength is for extended area sources (e.g., urban NOx emissions), and SO₂ comes from a collection of point sources (albeit occasionally in fairly close proximity), it is worthwhile to explore its effectiveness. Retrieving emissions on a grid smears out the emissions which usually can be geolocated to within a couple km, and makes getting the total for a facility sometimes challenging since one must figure out which grid boxes need to be summed over.*

The divergence method is fast and no model or other prior knowledge is needed for the emission derivation. The method does not need clearly isolated plumes, and it works also for areas with a mix of emissions (highways, towns, industry, power plants close together). The daily estimates from the divergence method and a longer time period of observations are needed. Up to now, the flux-divergence method has been proved to be effective for point source of $NO_x$ (Beirle et al., 2019), $CH_4$ (Liu et al., 2021) and $CO_2$ (Hakkarainen et al., 2022). In our study we explore the feasibility for $SO_2$ emissions and have discussed the spreading effect from the divergence method. One of our major efforts is to reduce the spreading caused by the divergence calculation. |With this improvement, the determination of the location of the $SO_2$ point sources improved.

*The authors attempt to calculate the lifetime considering simple physical (dry deposition) and first order chemical (via SO₂ + OH) loss, and these individual lifetime are combined. As an aside, why is only dry deposition included? Adding it would not impact my argument below, but it seems like wet deposition should be considered to be consistent with the approach chosen by the authors. Dry deposition lifetime were quite long, >60 hours, and thus do not have a huge impact on the combined lifetime. Chemical lifetime were calculated using an OH field (τ = 1/[k [OH]) from the ECMWF CAMS forecast model, and are roughly 20 hours, varying in space and time. The combined lifetime is something like 15 hours.*

In our study, we calculate the monthly mean $SO_2$ effective lifetime under the PBL height, as the major amount of $SO_2$ is concentrated inside the PBL. As a major oxidant in the troposphere (Crutzen, 1973; Logan et al., 1981), the OH concentration is high in India (Lelieveld et al., 2004; Lelieveld et al., 2016).Therefore, we assume the OH oxidation plays the most important role in determining the $SO_2$ lifetime. The $SO_2$ dry deposition, which occurs at the same time, has indeed a secondary influence on the lifetime (Chin et al., 1996). The $SO_2$ wet deposition

and other chemical reactions occurring in the cloud's droplets are less important (Smith and Jeffrey, 1975; Chin et al., 1996; Qu et al., 2019) compared to the OH oxidation and $SO_2$ dry deposition in terms of the monthly mean lifetime, especially since we use only cloud-free observations. In this case, we suggest that the $SO_2$ monthly mean lifetime will not change significantly even we involve the wet deposition and other chemical reactions. (See the answer of Point 2 below for more information.)

*My concerns regarding this lifetime issue are laid out here:*

*1.Using the OH field from ECMWF CAMS is not appropriate as the spatial resolution is 0.4 x 0.4 degrees (~40 km), and thus represent an average OH value, with half or more of the averaging area (on average; including the upwind portion) representing background values and chemistry. What is relevant is OH in the plume where the bulk of the $SO_2$ is, and OH at the plume core, plume edge, and background will all be much different. Using different models with comparable resolution to look at differences does not help in this regard. If one wants to use model OH then something like a plume-following chemical box model is the most appropriate choice.*

Just as the reviewer mentioned, the variation of the lifetime influencing factors should be considered. That is why we calculate the lifetime on each grid-cell. Solving the varied lifetime both in and out of the plume would be the way forward in the future. In this paper, we have made a first step by using CAMS output, which is indeed coarse but is already showing that concentrations and lifetimes can vary spatially and temporally and differ from simplified constant lifetime assumptions. We think it is a good start to derive the non-constant local lifetime for emission estimation.

Additionally, the OH changing within the plume is not fully ignored in our method although the resolution of the model grid is relatively coarse. To support this statement, we calculate the distance that is possible for $SO_2$ transportation according to the simple exponential decay function Eq. (1) and distance calculation function Eq. (2)

$$C = C_0 e^{-\frac{t}{\tau}}, \tag{1}$$

$$S = W\, t \tag{2}$$

In the given equations, $C$ represents the $SO_2$ concentration after it has decayed over a time $t$, given a lifetime $\tau$ and an initial concentration $C_0$. $S$ denotes the distance travelled by the plume

over the time $t$ at wind speed $W$. If we assume the initial concentration $C_0$ is 1 and $C$ is 10%, the possible distance at different wind speeds and lifetime conditions are listed below.

|          | $\tau=6h$       | $\tau=15h$      |
|----------|-----------------|-----------------|
| $W=1m/s$ | S=49.8 km       | S=124.3 km      |
| $W=2m/s$ | S=99.6 km       | S=248.7 km      |
| $W=3m/s$ | S=149.4 km      | S=373.0 km      |
| $W=4m/s$ | S=199.2 km      | S=497.4 km      |

Based on the table, we know that the $SO_2$ plume can cover more than two model grids with a resolution of 0.4° (about 40 km) even under very low wind conditions (2 m/s) and a short $SO_2$ lifetime (6 hours). Therefore, the OH change along the plume is considered on the resolution of our grid.

*2.In previous applications of the flux-divergence method, the lifetime was derived using the data itself (including and especially in the original formulation by Beirle et al. (2019) and this seems like the obvious method to employ here. Complications such as non-linear chemistry will then be accounted for. There is sufficient signal to tease out some spatial and seasonal differences from the TROPOMI data itself. At the very least the authors should have validated their calculated lifetimes using the data itself!*

In the background, without local emissions, we expect that the negative values from the flux divergence are cancelled out by the positive lifetime term. Therefore, the lifetime is optimised when this cancellation occurs in the divergence data. Figure S8 with fixed 6h lifetime seems to suggest a higher positive background as compared to Figure 6 with local, which indicates that the local lifetime is more realistic.

We further checked the averaged lifetime by deriving a new monthly mean $SO_2$ lifetime ($\bar{\tau}$) from the CAMS model by considering all $SO_2$ consuming processes and all kinds of sink according to Eq. (3),

$$\bar{\tau} = \frac{C}{E},$$ (3)

With $C$ being the $SO_2$ concentration and $E$ the $SO_2$ emission rate. We sum $M$ and $E$ for each month covering entire India to derive a monthly mean $\bar{\tau}$ averaged for the whole India. Fig. 2

shows the monthly $\bar{\tau}$ in 2019-2020 and 2022-2023 based on the CAMS model. The lowest lifetime is in summer, around 9.5 hours on average, while the longest lifetime is in winter, around 25.5 hours on average. The lifetime in spring and autumn is comparable, around 19 hours on average. The noticeable monthly/seasonal variation of lifetime align well with our calculations based on the OH oxidation and $SO_2$ dry deposition. This indicates that our calculated $SO_2$ lifetime will not change significantly even if wet deposition and other chemical reactions are considered now. Additionally, it is important to note that this model-intrinsic $SO_2$ lifetime of each month consistently exceeds 7 hours. It is evidence that a fixed lifetime is not the best choice for the emission estimation for the whole India.

In addition, we set a closed loop validation to validate the calculated $SO_2$ lifetime from CAMS model in Sec. 3.4 in the paper. This validation shows the emissions derived from the $SO_2$ lifetimes are comparable with the model input emissions, indicating that the lifetime we derive are in line with the model. Note that we account for the high uncertainty of the lifetime in the estimated error of the final emissions.

We add section 3.2.4 in the revised manuscript:

The monthly mean $SO_2$ effective lifetime is calculated based on OH oxidation and the $SO_2$ dry deposition. We assume negligible influence on lifetime from $SO_2$ wet deposition and other chemical reactions occurring in the cloud's droplets in terms of monthly mean lifetime, especially since we use only cloud-free observations. To show this, we derive a new monthly mean $SO_2$ lifetime ($\bar{\tau}$) from the CAMS model by considering all $SO_2$ producing processes and all kinds of sink according to Eq. (6),

$$\bar{\tau} = \frac{C}{E}, \tag{6}$$

with $C$ being the total $SO_2$ concentration and $E$ the total $SO_2$ emissions. We sum both the concentrations and the emissions of the model for each month covering the entire India to derive a monthly mean averaged $C$ and $E$ for the whole India. Fig. 2 shows the monthly $\bar{\tau}$ in 2019-2020 and 2022-2023 based on the CAMS model. This model-intrinsic $SO_2$ lifetime of each month consistently exceeds 7 hours. The lowest lifetime is in summer, around 9.5 hours on average, while the longest lifetime is in winter, around 25.5 hours on average. The lifetime in spring and autumn is comparable, around 19 hours on average. Note that the CAMS model

includes both dry and wet deposition of SO₂. The noticeable monthly/seasonal variation of lifetime align well with our calculations based on the OH oxidation and SO₂ dry deposition, indicating our calculated SO₂ lifetime will not change significantly even if wet deposition and other chemical reactions are considered. At the same time, we see a large variation both spatially as in the average from month to month. Therefore, we will use the monthly-averaged local lifetime from here on.

[Figure]

**Figure 2. Monthly averaged SO₂ lifetime in India for (a) 2019-2020 and (b) 2022-2023. The lifetime is calculated by accounting for all SO₂ producing processes and all kinds of sink in the CAMS model.**

*3.A simple analysis of the plumes themselves in the satellite data suggests that the effective lifetime is shorter than 15 hours. The authors do not ever show the actual TROPOMI SO2 data in their paper (there is one panel in the supplement), which seems strange considering it is the basis for the emissions calculations. Shown below in Figure 1 is an OMI SO2 VCD averaged over 2014-2017 (this is all I had handy; TROPOMI will look similar, but hot spots will appear sharper due to its higher spatial resolution). The dots are the larger SO2 sources. These are unpublished, diagnostic figures from the same EMG (exponentially modified Gaussian) method as published in Fioletov et al. (2016, 2018), McLinden et al. (2016) and later papers by that group. The left panel is the mean VCD (minus a slowly varying background bias, an artefact of the method, as discussed in the references above). The right is the reconstruction of the satellite data assuming a 6 hour effective lifetime.*

The TROPOMI $SO_2$ observations are shown in the new Figure 9b of the updated manuscript.

It is important to mention the purpose of this study, which is to derive a $SO_2$ emission inventory covering the whole India. The lifetime we use is the effective lifetime averaged for each grid cell of the area instead of averaged within the plume of the individual source. The EMG method is also relying on an effective lifetime since the lifetime within the plume is considered constant and averaged over a certain time period. Both effective lifetimes are not necessarily defined in the same way and both are approximations of the real lifetime.

In the EMG method the effective lifetime is derived under the assumption that the $SO_2$ lifetime is constant within a $SO_2$ plume (Fioletov et al., 2015). However, the factors influencing the $SO_2$ lifetimes, such as OH concentration, can vary significantly along the plume. For example, Krol et al. (2024) found OH concentration is lowest near the source and gradually increase with distance. They found the lifetime of $NO_2$ and $NO_x$ around the studied coal-fired power station are longest near the source and gradually shorten with increasing distance. Although this study focuses on $NO_x$ instead of $SO_2$, it provides insights that the lifetime varies spatially due to the spatial variation of influencing factors. Since both $NO_2$ and $SO_2$ are converted through reactions with OH, this study provides information also relevant for $SO_2$.

*4.For EMG methods similar to that employed in my point 3, de Foy et al. (2015) argued that there is a dispersion component to the effective lifetime, and it reduces the overall lifetime relative to the chemical+physical lifetime. In my understanding of the flux divergence method, diffusion is not accounted for and so it seems reasonable that the effective lifetime appropriate*

*for this method may also have a dispersive component. Further, in the case of an isolated point source, as it the case for many here, the two methods are virtually the same with the main difference being that the EMG method of Fioletov et al. assumes a plume shape, and the flux divergence method does not constrain the plume, but derives emissions on a grid. If this case, the physical interpretation and value of the effective lifetime should be very similar, and I would argue that my Figure 1 above is the most compelling argument for a shorter lifetime. See Vindhyachal in particular (the most northerly isolated hotspot). If the physical interpretation of the lifetime is not the same, then one would need to look at how the different methods perform when emissions are well know, such as from CEMS or other direct measurements. The EMG method of Fioletov et al has been extensively validated against such measurements in the US, Europe, and for volcanic sources as well.*

The divergence method is based on local mass balance and therefore it includes diffusion. The dispersive component in the divergence method leads to spreading of the signal (noise on the divergence) but it doesn't affect its lifetime calculation. By averaging over a longer time period, the effect is strongly reduced. In future studies we hope also to have results for the US or Europe, allowing a more extensive validation for point sources.

*5.Application to India is more difficult since there is little reference data to validate against. In such cases it is useful to apply the method to other locations where such data is available. For example, in the US, there is CEMS data to compare against. The authors even mention it can applied anywhere. If one wants their results to be received with confidence, this is an essential step.*

The high and evenly distributed $SO_2$ point-source emissions in India (Fioletov et al., 2023) provides an ideal base for developing and applying the divergence method. This is because the divergence method particularly works well for identifying strong point-source emissions, like Indian $SO_2$ emissions. We also did the test for Europe and US. The emission maps of these clean regions mix with more noise compared to the India case, making it not a good a base for improving the divergence method.

The Indian high but not clear $SO_2$ emissions itself is also an important reason why we focus on India. The Indian $SO_2$ emissions change rapidly. This study aims to provide an updated $SO_2$

budget for India for recent years. In future research, we certainly would like to also focus on regions where more reference data is available, like China.

*Additional comments:*

*Presumably there is nothing that unusual about Indian OH levels (the lifetime as derived here, to first order only depends on k and [OH]), and thus had the authors preformed a global analysis they would have found similar results (ie, ~a factor of two smaller emissions as compared to current, best emissions values) everywhere. Even just for India this has large implications, but for the globe this would lead us to rethink the $SO_2$ budget.*

In the recent ATMOS conference in 2024, we learned that the latest global top-down $SO_2$ emission estimation from Adrian Jost and Steffen Beirle in Max Planck Institute for Chemistry is 30% lower than the estimation from Fioletov et al. (2023). The online abstract might be available after August 30, 2024. In the EGU conference 2024, the Indian researcher Pramod Kumar working at Laboratoire des Sciences du Climat et l'Environnement, France estimates that Indian $SO_2$ emissions are approximately 5.0 Tg year$^{-1}$ in recent years. The work is now on writing. We notice the previous studies seldom provide conclusions for the present or future $SO_2$ emissions in India. And this study aims to provide more information of the potential Indian $SO_2$ emissions in recent years, like what the other researchers are also doing now. Since (1) the climate zone of India is very different from other regions in the world, (2) bottom-up emissions have a high uncertainty in India, and we have estimated a high uncertainty on our own emissions, we will not draw any conclusion about the $SO_2$ budget of the entire world.

*Line 382: "But we see a noticeable smoothing effect and an overall positive bias on emissions estimated with a fixed 6-hour lifetime compared to the emissions estimated with a local, variable lifetime, especially around the source location"*

*But given the colour-scale, and comparing Figure S8 with Figure 5, this is exactly what one would expect when you ~double the emissions... there should be more red because all values are larger for the 6 hour lifetime, and thus it would appear to be smoother. The only way this line of reasoning could be reliably used is to compare an isolated source, and then normalize them their peak value. Eyeballing Vindhyachal, the large, isolated point source in my Figure 2, below, the effect, even with the 2x emissions, appears marginal. Bigger picture, this type of analysis is indirect at best... again, why not use the actual data to deduce the lifetime?*

We have made efforts to ensure that the emission signal remains concentrated at the source rather than spreading to the surrounding areas. Therefore, after filtering the noise, we expect the emission signal can represent the point source emissions without large spreading. We have indeed considered the effect of the color scale. Therefore, as recommended by the reviewer, we have changed the color bar range in Fig. 6 and Fig. S8 in the revised manuscript by adapting the color range to the maximum value. The general positive bias still exists in the "6-hour" emission map after adapting the color bar (see the comparison figure of a zoom-in area below). The 6-hour lifetime is deduced from the actual measurement (Fioletov et al., 2015).

[Figure]

**Figure R1. SO$_2$ emissions in a zoom-in area with dense SO$_2$ point sources in India. (a) is the same as Fig. 6d in the paper. It represents the SO$_2$ emissions calculated based on non-constant SO$_2$ lifetime (b) the same is Fig. S8d in the supplementary information. It represents the SO$_2$ emissions calculated based on a constant 6-hour lifetime.**

**Reviewer 2,**

*This paper uses a flux-divergence method to estimate monthly SO₂ emissions in India based on satellite observations. The paper is well organized but I have a little concern about the calculation of SO₂ lifetimes, which is crucial to the final emission estimation. The authors declaimed that the satellite overpasses at noon time and they used SO₂ under cloud-free conditions, so they considered only the gas phase loss of SO₂. It may be true that SO₂ is mainly removed through the reaction with OH as the satellite overpasses, but SO₂ lifetime is as long as tens of hours, during this time, SO₂ is removed mostly by the liquid phase or the heterogeneous reactions, both of which is faster and more efficient than the gas phase reaction. So I'm not sure if it is appropriate to consider only gas phase loss of SO₂ in the SO₂ lifetime estimation.*

In our study, we calculate the monthly mean $SO_2$ effective lifetime under the PBL height, as the major amount of $SO_2$ is concentrated below the PBL. As a major oxidant in troposphere (Crutzen, 1973; Logan et al., 1981), the OH concentration is high in India (Lelieveld et al., 2004; Lelieveld et al., 2016).We therefore assume the OH oxidation plays the most important role in determining the $SO_2$ lifetime. The $SO_2$ dry deposition, which occurs at the same time, has indeed a secondary influence on the lifetime (Chin et al., 1996). The $SO_2$ wet deposition and other chemical reactions occurring in the cloud's droplets are less important (Smith and Jeffrey, 1975; Chin et al., 1996; Qu et al., 2019) compared to the OH oxidation and $SO_2$ dry deposition in terms of the monthly mean lifetime, especially since we use only cloud-free observations. In this case, we suggest that the $SO_2$ monthly mean lifetime will not change significantly even we involve the wet deposition and other chemical reactions.

To show this, we derive a new monthly mean $SO_2$ lifetime ($\bar{\tau}$) from the CAMS model by considering all $SO_2$ producing processes and all kinds of sink according to Eq. 1,

$$\bar{\tau} = \frac{C}{E},$$
(1)

With $C$ being the $SO_2$ air mass and $E$ the $SO_2$ emission rate. We sum $C$ and $E$ for each month covering the entire India to derive a monthly mean $\tau$ averaged for the whole India. Fig. 2 shows the monthly $\bar{\tau}$ in 2019-2020 and 2022-2023 based on the CAMS model. The lowest lifetime is in summer, around 9.5 hours on average, while the longest lifetime is in winter, around 25.5

hours on average. The lifetime in spring and autumn is comparable, around 19 hours on average. Note that the CAMS model includes both dry and wet deposition of $SO_2$. The noticeable monthly/seasonal variation of lifetime align well with our calculations based on the OH oxidation and $SO_2$ dry deposition. This indicates that our calculated $SO_2$ lifetime will not change significantly even if wet deposition and other chemical reactions are considered. Additionally, it is important to note that this model-intrinsic $SO_2$ lifetime of each month consistently exceeds 7 hours. The lifetime we derived here and shown in the paper also match the results ranging in 0.6-2.6 days of previous studies (Lee et al., 2011 and the papers therein).

In addition, we set a closed loop validation to validate the calculated $SO_2$ lifetime from CAMS model in Sec. 3.4 in paper. This validation shows the emissions derived from the $SO_2$ lifetimes are comparable with the model input emissions, indicating the lifetimes we derive are in line with the model. We also include the high uncertainty of these lifetimes in the estimated error of the final emissions.

We add section 3.2.4 started from line 231 in the revised manuscript:

The monthly mean $SO_2$ effective lifetime is calculated based on OH oxidation and the $SO_2$ dry deposition. We assume little influence on the lifetime from $SO_2$ wet deposition and other chemical reactions occurring in the cloud's droplets in terms of monthly mean lifetime, especially since we use only cloud-free observations. To show this, we derive a new monthly mean $SO_2$ lifetime ($\bar{\tau}$) from the CAMS model by considering all $SO_2$ consuming processes and all kinds of sink according to Eq. (6)

$$\bar{\tau} = \frac{C}{E}, \tag{6}$$

with $C$ being the total $SO_2$ concentration and $E$ the total $SO_2$ emissions. We sum both the concentrations and the emissions of the model for each month covering the entire India to derive a monthly mean averaged $C$ and $E$ for the whole India. Fig. 2 shows the monthly $\bar{\tau}$ in 2019-2020 and 2022-2023 based on the CAMS model. This model-intrinsic $SO_2$ lifetime of each month consistently exceeds 7 hours. The lowest lifetime is in summer, around 9.5 hours on average, while the longest lifetime is in winter, around 25.5 hours on average. The lifetime in spring and autumn is comparable, around 19 hours on average. Note that the CAMS model

includes both dry and wet deposition of SO₂. The noticeable monthly/seasonal variation of lifetime align well with our calculations based on the OH oxidation and SO₂ dry deposition, indicating our calculated SO₂ lifetime will not change significantly even if wet deposition and other chemical reactions are considered. At the same time, we see a large variation both spatially as in the average from month to month. Therefore, we will use the monthly-averaged local lifetime from here on.

[Figure]

**Figure 2. Monthly averaged SO₂ lifetime in India for (a) 2019-2020 and (b) 2022-2023. The lifetime is calculated by accounting for all SO₂ producing processes and all kinds of sink in the CAMS model.**

*The SO$_2$ emission estimated here is much less (even 50%) than other datasets, does this indicate that the SO$_2$ lifetime calculated in this study is too long?*

Since our resulting emissions are half as low as those of some other studies, we provide additional evidence to support our results in the revised manuscript. In the bottom-up emission inventories, i.e. CAMS-GLOB-ANT and EDGAR, the estimation of Indian SO$_2$ anthropogenic emissions is in the range of 10-14 Tg/year. It is calculated that India emitted approximately 11 Tg/year according to the CAMS-GLOB-ANT v5.3 inventory in 2023. However, we found the CAMS model simulated SO$_2$ densities, which are driven by the CAMS-GLOB-ANT v5.3, are much higher by a factor of 2 than TROPOMI SO$_2$ measurements in recent years (see the comparison in 2023 in Fig. 9). Considering the good quality of the TROPOMI observations (Theys et al., 2021), we attribute the higher simulation results to the overestimation of the model input emissions.

We add the discussion started from line from 445 in the revised manuscript:

We calculate Indian SO$_2$ emissions to be 5.2 Tg year$^{-1}$ using the SO$_2$ local lifetime, and 12.0 Tg year$^{-1}$ using a fixed 6-hour lifetime. The country-total emission obtained with a local lifetime are about 50% lower than the reported emissions in the most used bottom-up inventories, i.e. CAMS-GLOB-ANT and EDGAR. The CAMS-GLOB-ANT v5.3 inventory estimates that India emitted 11 Tg year$^{-1}$ SO$_2$ in 2023. However, the CAMS model simulated SO$_2$ densities, driven by CAMS-GLOB-ANT v5.3, are much higher by a factor of 2 than the TROPOMI measurements (see the comparison for 2023 in Fig. 9). Considering the good data quality of the TROPOMI observations (Theys et al., 2021; De Smedt et al., 2021) and assuming the CAMS model has a good performance, we attribute the higher simulation results mainly to a positive bias in the emissions that were input to the model.

[Figure]

**Figure 9. Indian SO₂ vertical column densities (VCDs) averaged in 2023 from (a) CAMS global composition forecast dataset, and (b) TROPOMI Level-2 COBRA dataset (at about the overpass time of 6 UTC). We integrate the TROPOMI observations to a resolution of 0.4° × 0.4°, the same as the CAMS datasets. (c) is the difference obtained by subtracting (b) from (a). The data of the same days are used for comparison The CAMS SO₂ density with total cloud coverage larger than 30% are excluded from the averaging.**

*And a minor comment, line 213: 'the SO₂ monthly dry deposition lifetime within the PBL height is calculated by dividing the PBL height by 0.4 cm s⁻¹', shouldn't it be '… dividing half the PBL height by 0.4 cm s⁻¹'?*

Slinn et al. (1978) calculated the dry deposition lifetime within a layer by dividing the height of the layer by the dry deposition velocity. Similarly, we calculate the dry deposition lifetime within PBL by dividing the PBL height by the velocity of 0.4cm s⁻¹. We treat the PBL as a single layer here.

**Reference**

Beirle, S., Borger, C., Dörner, S., Li, A., Hu, Z., Liu, F., Wang, Y., and Wagner, T.: Pinpointing nitrogen oxide emissions from space, Sci. Adv., 5, eaax9800, doi:10.1126/sciadv.aax9800, 2019.

Chin, M., Jacob, D. J., Gardner, G. M., Foreman-Fowler, M. S., Spiro, P. A., and Savoie, D. L.: A global three-dimensional model of tropospheric sulfate, J. Geophys. Res. Atmos., 101, 18667-18690, https://doi.org/10.1029/96JD01221, 1996.

Crutzen, P.: A discussion of the chemistry of some minor constituents in the stratosphere and troposphere, PApGe, 106, 1385-1399, 10.1007/BF00881092, 1973.

De Smedt, I., Pinardi, G., Vigouroux, C., Compernolle, S., Bais, A., Benavent, N., Boersma, F., Chan, K. L., Donner, S., Eichmann, K. U., Hedelt, P., Hendrick, F., Irie, H., Kumar, V., Lambert, J. C., Langerock, B., Lerot, C., Liu, C., Loyola, D., Piters, A., Richter, A., Rivera Cárdenas, C., Romahn, F., Ryan, R. G., Sinha, V., Theys, N., Vlietinck, J., Wagner, T., Wang, T., Yu, H., and Van Roozendael, M.: Comparative assessment of TROPOMI and OMI formaldehyde observations and validation against MAX-DOAS network column measurements, Atmos. Chem. Phys., 21, 12561-12593, 10.5194/acp-21-12561-2021, 2021.

Fioletov, V. E., McLinden, C. A., Krotkov, N., and Li, C.: Lifetimes and emissions of SO2 from point sources estimated from OMI, Geophys. Res. Lett., 42, 1969-1976, 10.1002/2015gl063148, 2015.

Fioletov, V. E., McLinden, C. A., Griffin, D., Abboud, I., Krotkov, N., Leonard, P. J. T., Li, C., Joiner, J., Theys, N., and Carn, S.: Version 2 of the global catalogue of large anthropogenic and volcanic SO2 sources and emissions derived from satellite measurements, Earth Syst. Sci. Data, 15, 75-93, 10.5194/essd-15-75-2023, 2023.

Hakkarainen, J., Ialongo, I., Koene, E., Szelag, M. E., Tamminen, J., Kuhlmann, G., and Brunner, D.: Analyzing Local Carbon Dioxide and Nitrogen Oxide Emissions From Space Using the Divergence Method: An Application to the Synthetic SMARTCARB Dataset, FRONTIERS IN REMOTE SENSING, 3, 10.3389/frsen.2022.878731, 2022.

Krol, M., van Stratum, B., Anglou, I., and Boersma, K. F.: Estimating NOx emissions of stack plumes using a high-resolution atmospheric chemistry model and satellite-derived NO2 columns, EGUsphere, 2024, 1-32, 10.5194/egusphere-2023-2519, 2024.

Lelieveld, J., Dentener, F. J., Peters, W., and Krol, M. C.: On the role of hydroxyl radicals in the self-cleansing capacity of the troposphere, Atmos. Chem. Phys., 4, 2337-2344, 10.5194/acp-4-2337-2004, 2004.

Lelieveld, J., Gromov, S., Pozzer, A., and Taraborrelli, D.: Global tropospheric hydroxyl distribution, budget and reactivity, Atmos. Chem. Phys., 16, 12477-12493, 10.5194/acp-16-12477-2016, 2016.

Liu, M., van der A, R., van Weele, M., Eskes, H., Lu, X., Veefkind, P., de Laat, J., Kong, H., Wang, J., Sun, J., Ding, J., Zhao, Y., and Weng, H.: A New Divergence Method to Quantify

Methane Emissions Using Observations of Sentinel-5P TROPOMI, Geophys. Res. Lett., 48, e2021GL094151, https://doi.org/10.1029/2021GL094151, 2021.

Logan, J., Prather, M., Wofsy, S., and McElroy, M.: Tropospheric chemistry: A global perspective, Journal of Geophysical Research, 86, 10.1029/JC086iC08p07210, 1981.

Qu, Z., Henze, D. K., Li, C., Theys, N., Wang, Y., Wang, J., Wang, W., Han, J., Shim, C., Dickerson, R. R., and Ren, X.: SO2 Emission Estimates Using OMI SO2 Retrievals for 2005–2017, J. Geophys. Res. Atmos., 124, 8336-8359, https://doi.org/10.1029/2019JD030243, 2019.

Slinn, W. G. N., Hasse, L., Hicks, B. B., Hogan, A. W., Lal, D., Liss, P. S., Munnich, K. O., Sehmel, G. A., and Vittori, O.: Some aspects of the transfer of atmospheric trace constituents past the air-sea interface, Atmospheric Environment (1967), 12, 2055-2087, https://doi.org/10.1016/0004-6981(78)90163-4, 1978.

Smith, F. B. and Jeffrey, G. H.: Airborne transport of sulphur dioxide from the U.K, Atmospheric Environment (1967), 9, 643-659, https://doi.org/10.1016/0004-6981(75)90008-6, 1975.

Theys, N., Fioletov, V., Li, C., De Smedt, I., Lerot, C., McLinden, C., Krotkov, N., Griffin, D., Clarisse, L., Hedelt, P., Loyola, D., Wagner, T., Kumar, V., Innes, A., Ribas, R., Hendrick, F., Vlietinck, J., Brenot, H., and Van Roozendael, M.: A sulfur dioxide Covariance-Based Retrieval Algorithm (COBRA): application to TROPOMI reveals new emission sources, Atmos. Chem. Phys., 21, 16727-16744, 10.5194/acp-21-16727-2021, 2021.

---

## Author Response (AR2)

Response to reviewer 3,

We thank the reviewer for spending time on the paper improvement. Below, we provide a point-by-point response to the comments, and we list the related changes made to the manuscript. The original review text is indicated in italic blue font and the response in regular black font. New text appearing in the revised paper is indicated by a red color. It is noted that the line number we mentioned here is for the manuscript with the track changes.

*The study "SO2 emissions and lifetimes derived from TROPOMI observations over India using a flux-divergence method" by Chen et al. presents emission estimates over India based on TROPOMI observations and a flux-divergence method. The paper generally reads well and presentation quality is good. However, I don't see the main conclusion to be sufficiently supported by the presented results, as detailed below. While I don't think that all the open questions need to be solved within this study before publication, the authors need to extend the discussion substantially and have to present their conclusions more cautious and less absolute.*

The main aim in this paper is to update the methodology typically used for deriving $SO_2$ emissions from satellite data by introducing a lifetime for $SO_2$ in the inversion calculation. In previous studies it was often considered constant, and thus independent of latitude or season. By calculating a seasonal and latitude dependence into the local $SO_2$ lifetime, we investigate the effect on the resulting emission estimates. This paper is considered a first step towards addressing the lifetime variability in the inversion methodology.
We have added the text to the manuscript in the last paragraph of the paper to indicate that further work should be done:
For those regions with more Northerly latitudes than 40°N (e.g. Northern China, Eastern Europe), the latitude and season dependent $SO_2$ lifetime with the improved divergence approach has the potential to significantly improve the top-down derivation of $SO_2$ emission estimates. This paper is considered a first step towards addressing the lifetime variability in the inversion methodology.

*Before publication, the following issues need to be resolved:*

*Major issues:*

*1. Title*
*SO2 emissions have been derived from TROPOMI observations.*
*However, the lifetimes have not! They are actually based on CAMS simulations, and the CAMS-based lifetimes are the basis of the conclusions that emissions in CAMS are too high.*
*Thus, the title is misleading, and "and lifetimes" should be skipped.*

Actually, our lifetimes are not directly based on CAMS simulations, but they are calculated based on previous studies while using an OH climatology based on CAMS. However, we agree that the lifetimes are

not based on TROPOMI observations either and the title has been changed to "SO$_2$ emissions derived from TROPOMI observations over India using a flux-divergence method with variable lifetimes.

**2. SO$_2$ lifetime**

*The main finding of the study is that TROPOMI SO$_2$ columns over India are lower than CAMS model simulations, and the conclusion of the study is that bottom-up emissions of SO$_2$ are too high. However, the discrepancy could also have other reasons, in particular that the modeled SO$_2$ lifetime is too high.*

*I see the following aspects that need further discussion:*

(a) *OH climatology: The SO$_2$ lifetime is estimated based on CAMS OH climatology. However, this approach completely ignores the very special conditions within the power plant plumes: In particular due to the co-emitted NOx, plume-OH is probably highly variable and can substantially differ from the climatology!*

We add the discussion of OH chemistry occurring within the NO$_x$ plume in Sec. 7 between line 503-510:

The hard-to-quantify factors influencing the lifetime and emissions are discussed here. First, our grid-averaged (about 40 km ×40 km) OH climatology does not resolve the detailed chemical variation within the pollutant plumes, particularly those involving the interaction between NO$_x$ and OH. Krol et al. (2024) studied the chemistry within the NO$_x$ plumes and observed low OH concentrations near the strong NO$_x$ sources (within an average of 10 km) and high OH away from the sources. This suggests that our 40 km averaged OH climatology cannot capture this OH decline and may underestimate the SO$_2$ lifetime near the large NO$_x$ sources. Furthermore, the variation of OH concentration between 10 km to 40 km from the source is roughly limited to 10 %, see Fig. 7b from Krol et al. (2024). We have considered these effects in our error estimate of the lifetime.

(b) *) Clear sky: The authors claim that only reaction with OH is relevant, as only cloud free observations were considered. However, this is not completely true:*
*- A cloud filter was applied, but pixels with up to 30% cloud fraction are still included! (as I understand from Fig. 9; the cloud threshold has to be added to section 2.1)*
*Even if a pixel is cloud free at TROPOMI overpass, the observed air mass might have been in contact with clouds over the last hours.*
*Also heterogenous reactions with aerosols can play a role.*

Analysis of the dominant term for conversion in the CAMS system shows aqueous phase chemistry is the dominant term related to sulphate production rather than heterogeneous reactions therefore we assume wet-aerosols are a negligible source term (which typically have low pH and slow sulphate production (Gillani et al., 1981)).

We have added the text between line 510 and line 519:

Second, we did not consider the heterogenous $SO_2$ reactions on wet aerosols. We suppose this impact on $SO_2$ lifetime can be neglected in our study. Analysis of the dominant term for conversion in the CAMS system shows aqueous phase chemistry is the dominant term related to sulphate production rather than heterogeneous reactions. We therefore assume wet aerosols are a negligible source term (which typically have low pH and slow sulphate production (Gillani et al., 1981)). Even though the atmospheric $SO_2$ in gas phase can convert to aqueous phase and be oxidized to form sulfate on aerosol wet surface or within clouds, these reactions typically occur on hazy days with high relative humidity and PM2.5 level (Ge et al., 2021). These meteorological conditions are generally not favored on days with minimal cloud coverage, as achieving the necessary high relative humidity is difficult with ample sunlight at noon.

TROPOMI can only "see" $SO_2$ from point sources located at the surface for cloud-free pixels. It means only cloudless area $SO_2$ can be measured by TROPOMI even though up to 30% clouds may exist. The TROPOMI retrievals for the whole pixel are based on the cloud-free part only. Therefore, we calculate an effective lifetime for cloud-free conditions. In theory, it might be possible that plumes originate under the cloud and later become visible in cloud-free pixels. We assume that these cases are rare and will only lead to a small additional uncertainty.

We added an extra explanation about the uncertainty of the $SO_2$ lifetime we described.
To emphasize that we only calculate the $SO_2$ lifetime in the cloudless area, we have added new text between line 188 and 190:

Notably, TROPOMI can "see" $SO_2$ only in the cloud-free part of the pixel, leaving $SO_2$ concentrations within or beneath clouds being unmeasurable. We assume that the resulting $SO_2$ has had no interaction with clouds, thus the resulting lifetime derived for $SO_2$ pertains to cloud-free conditions in a constrained region.

We add the text in Section 3.2.3 between line 233 and 236:

Comparing the $SO_2$ lifetime derived here with those proposed in the literature shows that our estimates are similar to other independent model-based lifetime estimates (Lee et al., 2011) and ground-measurement based lifetime estimates (Hains et al., 2008). We therefore argue that on average our calculated $SO_2$ lifetime is reasonable. Furthermore, it has a latitude and seasonal dependency that is often lacking in other inversion methods.

And we add extra discussion of clouds effect on lifetime in Section 7 between 519 and 521:

Finally, we only calculate the lifetime for $SO_2$ in cloud-free regions, excluding the $SO_2$ wet deposition and the reactions within the clouds. This is actually the lifetime we need in our inversion since the TROPOMI observations of $SO_2$ plumes are limited to cloud-free scenes.

*My strongest concern about the main conclusion is actually raised by Fig. 9: While the difference in SO₂ columns is clear, this could indeed be due to input emissions, but also due to the loss processes in CAMS. It is hard to tell from the Figure alone, but it looks like CAMS (even if values would be halved) shows substantial outflow of SO₂, e.g. over the oceans, that is not observed from TROPOMI. Thus I would conclude that the CAMS SO₂ lifetime is definitely too high.*

The OH concentration is the major controlling factor of $SO_2$ lifetime under cloud-free conditions. We firstly compare the Indian OH concentration in CAMS with those in previous studies. Hewitt and Harrison (1985) summarized the early papers and found the OH concentrations were typically measured or simulated at around $10^6$ radicals/cm³, which is comparable to CAMS OH levels. Then we found that Indian CAMS OH levels are similar to those in Lelieveld et al. (2004) study ($2.0$-$4.0 \times 10^6$ radicals/cm³) (Fig R1), while slightly higher than the values in Lelieveld et al. (2016) study ($1.2$-$1.8 \times 10^6$ radicals/cm³). We suggest this difference arises because Lelieveld et al. (2016) reported OH concentrations averaged throughout the troposphere, whereas our focus is specifically on PBL-averaged OH concentrations. We also change the color bar range of Fig. 9 to fit their own maximum values and show it here (Fig. R2). Many strong source signals are shown in west of India on the CAMS map while not being visible on the TROPOMI map. Therefore, we suggest that the discrepancies in Fig. 9 mainly resulted by the overestimation of model input emissions. But we have added the text to the manuscript between line 469 and 476 to make this conclusion less absolute (See the text below the figure).

[Figure]

**Figure R1. Left: Annual mean OH concentration from CAMS model averaged from 2019 to 2023 in our study. Right: Annual mean OH concentrations near the earth's surface calculated with a chemistry-transport model (Lelieveld et al., 2004). The units are $10^6$ radicals/cm³.**

[Figure]

**Figure R2. Indian SO₂ vertical column densities (VCDs) averaged in 2023 from (a) CAMS global composition forecast dataset, and (b) TROPOMI Level-2 COBRA dataset (at about the overpass time of 6 UTC). We integrate the TROPOMI observations to a resolution of 0.4° × 0.4°, the same as the CAMS datasets.**

The emissions at the large source locations show big differences between the two maps. Many strong source signals in the west of India are shown on the CAMS map while not visible on the TROPOMI map. Considering the low uncertainties of the TROPOMI observations (Theys et al., 2021; De Smedt et al., 2021), we suggest that the difference in Fig. 9 is primarily due to a positive bias in model input-emissions. It is noted that the SO₂ lifetime in CAMS may be overestimated and contributes to the higher simulated SO₂ concentration, even though the OH level, which mainly determines the SO₂ lifetime under cloud-free conditions, are similar between CAMS results (Fig. S10) and previous studies (Hewitt and Harrison, 1985; Lelieveld et al., 2004; Duncan et al., 2024)

*The authors need to*

*- include a discussion about plume chemistry (affected by strong local NOx emissions) on OH and how far it is appropriate to use the OH climatology here,*

*- be aware that "cloud free" does not really mean free of any clouds, and heterogeneous reactions on clouds and aerosols need to be discussed as well, which actually results in lower SO2 lifetimes,*

*- include a discussion about remaining uncertainties, and the possibility that the CAMS lifetime is actually too long, as indicated in Fig. 9.*

We already shown the text we added in the manuscript above. Considering the added text is mostly in Section 7, I put the whole Section 7 here for the convenience of reading.

**7. Discussion**

The hard-to-quantify factors influencing the lifetime and emissions are discussed here. First, our grid-averaged (about 40 km × 40 km) OH climatology does not resolve the detailed chemical variation within the pollutant plumes, particularly those involving the interaction between SO₂ and OH. Krol et al. (2024) studied the chemistry within the NOₓ plumes and observed low OH concentrations near the strong NOₓ sources (within an average of 10 km) and enhanced OH away from the sources. This suggests that our 40 km averaged OH climatology cannot

capture this OH decline and may underestimate the $SO_2$ lifetime near the large $NO_x$ sources. Furthermore, the variation of OH concentration between 10 km to 40 km is roughly limited to 10 % (See Fig. 7b from (Krol et al., 2024). We have considered these effects in our error estimate of the lifetime. Second, we did not consider the heterogenous $SO_2$ reactions on wet aerosols. We suppose this impact on SO2 lifetime can be neglected in our study. Analysis of the dominant term for conversion in the CAMS system shows aqueous phase chemistry is the dominant term related to sulphate production rather than heterogeneous reactions. We therefore assume wet aerosols are a negligible source term (which typically have low pH for this slow sulphate production). Even though the atmospheric $SO_2$ in gas phase can convert to aqueous phase and be oxidized to form sulfate on aerosol wet surface or within clouds, these reactions typically occur on hazy days with high relative humidity and PM2.5 level (Ge et al., 2021). These meteorological conditions are generally not favored on days with minimal cloud coverage, as achieving the necessary high relative humidity is difficult with ample sunlight at noon. Finally, we only calculate the lifetime for $SO_2$ in cloud-free regions, excluding the $SO_2$ wet deposition and the reactions within the clouds. This is actually the lifetime we need in our inversion since we only consider cloud-free scenes measured by TROPOMI.

*With these open questions, I consider the given uncertainties of 35% (monthly) and 10% (annually) to be far too low.*

We change the uncertainty to 40% for monthly $SO_2$ emissions and explain this in the new text between line 380 and 383:

Although the measured $SO_2$ plume has no interaction with the clouds during the TROPOMI overpass, the $SO_2$ may interact with clouds before and after this time to influence the effective $SO_2$ lifetime. Therefore, we take an uncertainty of 40%, which is larger than the averaged uncertainty (35%), for the derived monthly emissions,

*3. Implementation of the derivative*
*The authors state that the analysis was performed on 0.4° grid, on 0.1° grid, and "TROPOMI measured pixels". It is not clear to me what the latter actually means:*
*Is the analysis done on a regular grid with 5.5 km × 3.5 km resolution (as indicated in line 267)?*
*Or is the original "grid" of the TROPOMI measurements (along x across) used for calculating the derivative?*
*I would not understand the first option, and I see problems with the pixel size changing across the swath.*
*If the second option is meant, this should be explained and motivated in more detail, and a reference to de Foy and Schauer, 2022, who proposed this approach, has to be added.*

We conduct the divergence calculation on original TROPOMI pixels (along · cross, about 5.5 km×3.5 km at nadir). Because this is the finest resolution that we can use to calculate the divergence for TROPOMI data. Although the final emissions are on the same grid size of 0.1°, the divergence calculated on original TROPOMI pixels performs better than the one calculated directly on 0.1° regular grids. Thus, our emission is derived based on the divergence map with the second option. Overall, we suppose the emission map resolution can be improved by enhancing the grid size but will be finally limited by the pixel scale of TROPOMI.

To make it clear, we make the changes in the manuscript between line 279 to 281:

Original sentence: The divergence can also be calculated based on the TROPOMI measured pixels (5.5 km× 3.5 km) and later integrated to the regular grid cells of 0.1°× 0.1°.

Revised sentence: Since the emission map resolution is limited by the pixel scale of TROPOMI, we also calculate the divergence based on the original TROPOMI measured pixels (on an along × across track grid, about 5.5 km× 3.5 km at nadir varying with the viewing angle) and later integrated to the regular grid cells of 0.1°× 0.1° (de Foy and Schauer, 2022).

*The information content of the divergence term comes from the change of horizontal flux from one pixel to the next.*
*For this, a resolution of 0.4° is far too coarse.*
*I would expect that (a) calculating the divergence on the TROPOMI grid and (b) re-gridding and averaging it afterwards on high spatial resolution (0.05°) yields high-resolution information about the location of SO2 point sources, without the need for modifying the difference quotient.*

We use the resolution of 0.4° for the closed-loop validation only. The emissions calculated from the observations are in 0.1° and original TROPOMI pixel scales. Calculating $SO_2$ emissions at different scales reveals that increasing resolution can reduce but not eliminate the spreading effect inherent in the divergence method. Using different scaling factors further mitigate the effect based on the given resolution. We highlight the limitations of the current divergence method and introduce an improved approach that enable other users to enhance emission map resolution and quality without requiring higher resolution and, consequently, computational cost. Based on the tests we had, we thought the resolution of 0.1° is suitable. There are about 50 out of 450 pixels larger than 5.5×10 km in a line of across-track pixels. Since the TROPOMI track shifts daily and divergence is averaged seasonally, each regular cell captures divergence from both big and small pixels. This can mitigate the error introduced by integrating divergence from large pixels (>0.1°) into a 0.1° grid. In future studies, we will constrain $SO_2$ emissions in a finer resolution, e.g. 0.05°.

[Figure]

Figure. R3. The across-track ground pixel size of TROPOMI. On average the pixel size across-track is about 6 km.

We change the sentence between line 123 and 125 to:

The spatial resolution for the center of the swath is approximately 5.5 km × 3.5 km (7 km × 3.5 km before August 6, 2019) in nadir, and 5.5 km × 6 km on average over the swath.

*Additional comments:*

*Line 103: "and its divergence": "its" refers to the sink term, whereas the divergence is derived from the horizontal flux.*

Original sentence: The flux-divergence method, i.e. adding the independently derived $SO_2$ sink term and its divergence to obtain local emissions, is used for the emission estimation.

Revised sentence: The flux-divergence method, i.e. combining the independently derived $SO_2$ sink and divergence, is used to obtain local emissions.

*Line 127: Please specify the version of the COBRA $SO_2$ product.*

We use the COBRA dataset v01.00.01 and it is added in the manuscript.

*Line 349: This pixel size is valid for nadir. Towards the swath edges, pixels become significantly larger (even larger than 0.1°!)*

Most TROPOMI pixels are finer than 0.1°, with about 50 out of 450 pixels per across-track line exceeding 5.5 × 10 km. To make the sentence more precise, we changed the original sentence in the manuscript and show it below.

[revised manuscript text omitted]